# An experimental test of whether financial incentives constitute undue inducement in decision-making

**Sandro Ambuehl** ✉

Around the world, laws limit the incentives that can be paid for transactions such as human research participation, egg donation or gestational surrogacy. A key reason is concerns about 'undue inducement'—the influential but empirically untested hypothesis that incentives can cause harm by distorting individual decision-making. Here I present two experiments ($n = 671$ and $n = 406$), including one based on a highly visceral transaction (eating insects). Incentives caused biased information search—participants offered a higher incentive to comply more often sought encouragement to do so. However, I demonstrate theoretically that such behaviour does not prove that incentives have harmful effects; it is consistent with Bayesian rationality. Empirically, although a substantial minority of participants made bad decisions, incentives did not magnify them in a way that would suggest allowing a transaction but capping incentives. Under the conditions of this experiment, there was no evidence that higher incentives could undermine welfare for transactions that are permissible at low incentives.

Around the world, laws and guidelines limit the incentives that can be paid for transactions such as human research participation, surrogate motherhood, human egg donation and organ donation. These transactions are often legal if no or limited amounts of money change hands. While there are a number of complex ethical issues at play (for example, objectification[1] and distributive justice[2,3]), an important reason for these limits on incentives is the influential albeit conceptually vague and empirically largely untested notion of undue inducement[4]. This term describes the belief that participation incentives can harm participants, because "something is being offered that is alluring to the point that it clouds rational judgment…Attention is fixated on the benefit, disallowing proper consideration of the risks"[5]. The American Society for Reproductive Medicine, for instance, writes that "payments to women providing oocytes should be…not so substantial that they… lead donors to discount risks" and conjectures that "the higher the payment, the greater the possibility that women will discount risks"[6]. The National Bioethics Advisory Commission limits incentives that can be offered for participation in clinical trials on the basis of the assertion

that "offers of large sums of money…could lead some prospective participants to enroll…when it might be against their better judgment"[7]. Two distinct claims make up the undue inducement hypothesis (UIH). The positive part (UIH-positive) is a behavioural prediction. It posits that incentives cause participants to engage in biased information processing and motivated reasoning about the transaction. Indeed, some bioethicists explicitly describe undue inducement as "a cognitive distortion relating to the assessment of risks and benefits"[8]. The normative part (UIH-normative) is the composite claim that these changes cause harm and that some transactions are therefore acceptable only at low but not at high incentives.

This paper empirically tests the cognitive underpinnings of the UIH. Empirical progress on the issue is crucial. Restrictions are well advised if incentives really impede decision quality to an extent that outweighs the benefits of the increased payment (including, potentially, benefits to society and other parties). If, however, the effect of incentives on decision quality is minor, then limits on incentives simply amount to underpaying suppliers such as clinical trial participants,

Department of Economics and UBS Center for Economics in Society, University of Zurich, Zürich, Switzerland. ✉e-mail: sandro.ambuehl@econ.uzh.ch

surrogate mothers and organ donors for their valuable contributions. Such underpayment constrains supply and may thus impede medical progress, cause grief to aspiring parents with reproductive health issues and lead to preventable deaths from kidney failure. While incentives cannot harm rational decision makers, evidence of questionable decision-making is widespread[9]. Direct empirical evidence on the UIH is limited to a small number of case studies and unincentivized surveys concerning clinical trial participation[10–13]. While these provide suggestive evidence that payments do not alter judgements about study risks, they neither study real choices nor perform formal welfare analysis. The literature on motivated reasoning[14] does not directly speak to the UIH because it typically considers settings in which the bias benefits rather than harms the decision makers—for instance, by allowing them to maintain a positive self-image while dodging costly social obligations[15].

I investigated the UIH using laboratory experiments in which subjects' decisions were carried out. This is possible even with stakes on the order of dozens or hundreds of dollars for two reasons. First, as I will demonstrate, even these stakes cause the information biasing that UIH-positive predicts. The ethics literature views such biasing as a key cause of UIH-normative. Second, as the ethics literature clarifies, an offer that is genuinely too good to refuse cannot constitute undue inducement. Undue inducement requires that a participant with undistorted judgement would refuse the offer, but that the incentive warps participants' judgement in a way that makes them accept it[16]. Undue inducement therefore requires that an offer is high relative to the incentivized activity—but not so high that a person with undistorted judgement would accept it. I satisfy this requirement by choosing activities whose downsides are commensurate to the incentive amounts I employ. While the main contribution of this paper concerns general cognitive mechanisms, college students (my study population) are a typical target demographic for recruiting human egg donors and clinical trial participants.

The paper consists of two experiments and a theoretical model. Experiment 1 features a setting in which participants needed to evaluate the non-monetary consequences of a transaction in terms of dollars and cents. To give the UIH the best chance, I sought a visceral and aversive transaction that was unfamiliar to the participants and that allowed for biased information acquisition. Hence, I incentivized the participants to eat whole insects such as mealworms, silkworm pupae and mole crickets. Some participants learned that they would receive US$3 in exchange for eating bugs. Others learned that their incentive for the same transaction was US$30. To prevent the incentive from acting as a signal about the discomfort of eating the insects, the participants knew of both incentive amounts and that they were randomly assigned to one of them. Next, the participants chose between two videos to help them decide whether to participate in the transaction. They were titled 'Why you may want to try eating insects' and 'Why you may not want to try eating insects'.

As UIH-positive predicts, I found that a higher incentive increased demand for the encouraging video at the cost of the discouraging video. A comparison with a control condition that precluded access to the videos shows that they causally affected participation decisions. These findings dovetail with the idea that "payments lead donors to discount risks"[6].

UIH-normative predicts that this mechanism links higher incentives to lower welfare. I tested this prediction using the informed consumer paradigm, a dominant approach in behavioural welfare economics[17–19]. Initially, the participants decided whether to accept the transaction at the promised incentive solely on the basis of the selected video and a brief verbal description of the insects. After making that decision, the participants received the items. They had to view, touch and (inadvertently) smell them. On the basis of this substantial additional information, the participants then revealed the least amount of money they would need to swallow these insects (reservation price). For participants who accepted the transaction, the difference between the incentive they received and this reservation price reveals potential choice mistakes. If they committed to eating the insects for US$30 but found, after inspection, that they would need at least US$45 as compensation for their displeasure, they had inflicted harm of US$(45 − 30) = US$15 onto themselves. Generally, I define 'ex-post surplus' as the difference between the reservation price and the incentive if the participant accepted the transaction and zero otherwise. To test UIH-normative formally, I aggregated surplus across participants. I weighted negative values (losses) $\frac{\alpha}{1-\alpha}$ times as much as positive values (gains), for arbitrary values $\alpha \in (0, 1)$. Even though 10% to 20% of participants were harmed by their decision to participate, I found that it was never optimal to allow the transaction at the low incentive but prevent it at the high incentive, for any weight on losses—contrary to UIH-normative. (High weights on losses called for preventing the transaction altogether.) This result is robust to controlling for the possibility that the initial incentive exerted anchoring effects on reservation prices and for noisy elicitation of reservation prices.

Experiment 2 used a broader range of incentive amounts, permitted alternative welfare benchmarks and explored moderators of the effect of incentives on decision quality. The participants decided whether to accept a given payment of €20 to €80 in exchange for risking the loss of €100. After learning the incentive amount but before committing to a decision, the participants chose between advisors who provided informative but imperfect recommendations about what to do. A 'Bold Advisor' was likely to recommend participation. He always did so if the loss would not materialize, but he sometimes also did so if it would. A 'Cautious Advisor' was likely to recommend non-participation. He always did so if the risk would realize, but he sometimes also did so if it would not. Upon observing a recommendation from the chosen advisor, the participants decided whether to accept the transaction.

UIH-positive predicts that higher incentives increase the preference for the Bold Advisor and that this information biasing causally affects participation decisions. The data confirmed both these predictions.

UIH-normative predicts that due to these behavioural changes, higher incentives lower welfare. To measure whether the decision was good given the information the participant had at the time, rather than in hindsight, I used the reframed decisions paradigm, the second dominant approach in behavioural welfare economics[20–22]. Specifically, I calculated how confident a perfectly rational participant would have been that he or she would not lose the €100 after receiving the chosen advisor's recommendation. Stage 2 of the experiment presented the participants with a lottery whose probability of losing €100 equalled a hypothetical perfectly rational agent's confidence, but which—unbeknownst to the participants—paralleled the decision they faced in stage 1 in all other ways. I measured what amount of money the participants considered just as good as that lottery (their certainty equivalent). A participant's decision to accept the transaction in stage 1 was a mistake if the participant would rather lose a given sure amount of money than play the corresponding stage-2 lottery. The magnitude of their certainty equivalent measures the severity of the error.

I found UIH-normative to be robustly violated. While mistakes were common, they did not substantially vary with the incentive. Depending on the weight assigned to losses, it was therefore optimal to either permit the transaction at all incentives or to prevent it altogether. But allowing the transaction at low but not at high incentives was always suboptimal. This was the case in treatments that varied the stakes by an order of magnitude, in treatments with a three-month delay between the receipt of the incentive payment and the realization of potential negative consequences, and in treatments in which the risk of losing the €100 ranged from 20% to 80%.

What explains the simultaneous confirmation of UIH-positive and refutation of UIH-normative in my experiments? To answer this question, I considered how perfectly rational decision makers would inform themselves about the potential consequences of a transaction

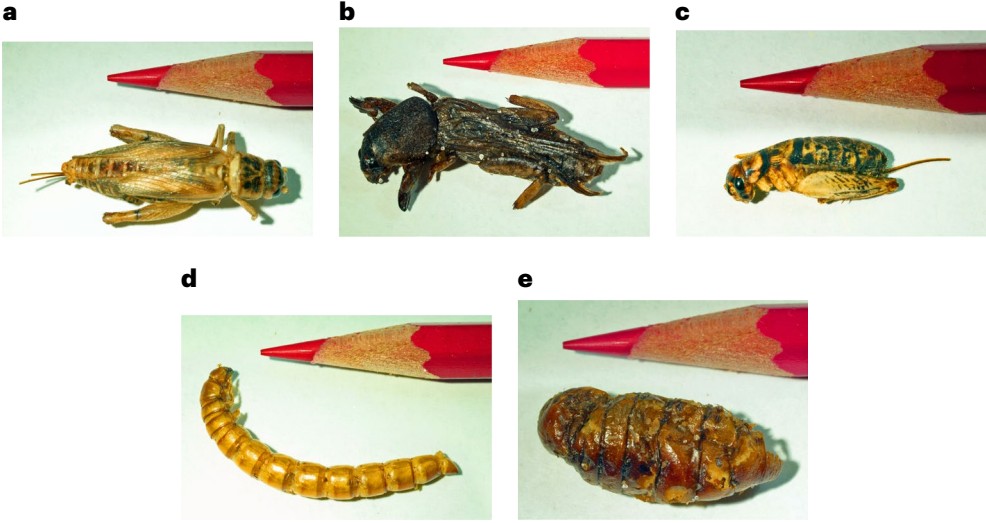

**Fig. 1 | Insects eaten by participants. a**, House cricket. **b**, Mole cricket. **c**, Field cricket. **d**, Mealworm. **e**, Silkworm pupa. The participants did not see the insects or pictures of them until after making participation decisions.

when doing so is costly. Because such decision makers will reject any transaction not in their interest, they will violate UIH-normative by construction. Yet, as I show, their information choice bears the hallmarks of UIH-positive: a higher incentive for participation lowers the cost of mistaken participation but increases the cost of mistaken abstention. Hence, rational agents will consult information sources they expect to be more likely to recommend participation and less likely to recommend abstention. The model thus shows that UIH-positive does not logically imply UIH-normative. While my experiment participants did not behave perfectly rationally, incentives changed the extent of their irrationality by sufficiently little to defy UIH-normative.

In addition to informing the debate on undue inducement, this paper contributes to three strands of academic literature. First, it adds to the literature on repugnant transactions, which studies third parties' judgements rather than the effect of incentive payments on the incentivized[23-30]. Second, this paper contributes to the literature in behavioural public economics[31] by examining the welfare effects of incentive payments—one of the most fundamental economic policy tools. Third, its welfare analysis of incentive-induced information biasing contributes to the literature on the positive test strategy[32] and motivated reasoning[14,33,34].

## Results

### Experiment 1: visceral transaction
Experiment 1 incentivized 671 US undergraduate participants to ingest real insects in exchange for money. This activity was novel and unfamiliar to most participants. Many found it intensely aversive. Some reported that the experiment was "stressful" or that the "insects were scary", while others refused even to touch the containers holding the dead animals.

**Design.** As described earlier, the participants were randomly assigned to receiving either US$3 or US$30 in exchange for eating insects and selected one of two six-minute videos that highlighted either the upsides or the downsides of eating insects. To provide more continuous data about information preferences, the participants also selected at least four of nine clips grouped in bins of three named 'Reasons for eating insects', 'Reasons against eating insects' and 'Other information about eating insects'. The participants knew that they would not receive any further information and would decide whether to eat the insects on the basis of only a brief verbal description. They thus had an incentive to select the videos carefully and to pay attention to their content.

With an exogenous 97% chance, the participants watched their selected video. Otherwise, they watched the selected clips.

Next, the participants revealed the least amount of money for which they would eat insects from each of five different species (their reservation prices) through a procedure that made the decision potentially consequential and rendered truth-telling optimal (Methods).

In the Main Decisions, which determined each participant's consumption with 80% probability, the participants then decided whether to eat insects from each of the five species in exchange for the incentive promised in the beginning. These were the decisions whose welfare consequences I sought to evaluate.

At this point, the participants had only seen brief verbal descriptions of the insects and the video they selected. The participants then received five containers filled with the insects they may have been be about to swallow (Fig. 1) and were forced to view, touch and smell each insect.

On the basis of this substantial additional information, the participants again revealed reservation prices for each species. I define a participant's ex-post surplus from such a decision as the difference between the reservation price and the incentive if the participant accepted the transaction, and zero otherwise. A negative value indicates that the Main Decision was mistaken in hindsight. Its magnitude is the severity of the error.

**Information choice and participation decisions.** To test UIH-positive, I examined how the incentive affected information choice. Figure 2a displays the fraction of participants opting for the encouraging video (left) and the number of positive and negative clips they selected (right), both as a function of the incentive amount. The US$30 incentive raised the proportion of participants who selected the encouraging video from 81.2% to 88.7%, a 7.5-percentage-point difference (t-test, $t = 2.09$, $P = 0.037$). Preferences for the video clips showed a similar effect. The difference between the numbers of positive and negative clips chosen rose from 0.86 to 1.31 (t-test, $t = 2.61$, $P = 0.009$) as the incentive increased. The chosen number of clips labelled 'other' remained approximately unchanged (t-test, $t = -0.03$, $P = 0.977$). Incentives biased information acquisition, consistent with UIH-positive.

Potentially harmful effects of incentives can arise only if these changes affect subsequent decisions. Yet, the effect of the incentive on information choice can also be explained by the hypothesis that participants decided whether to eat the insects before selecting a

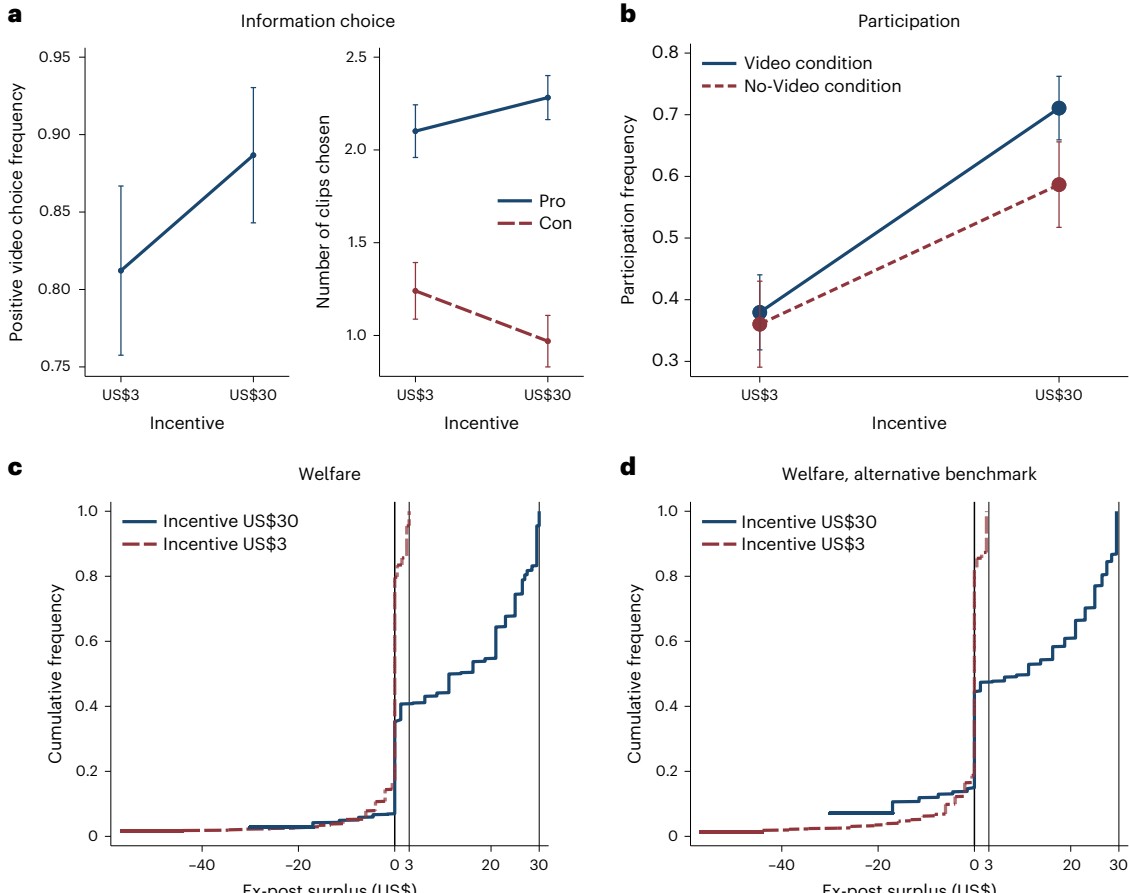

**Fig. 2 | Results from Experiment 1. a**, Information choice. Left, the fraction of participants who chose the encouraging video (400 independent participants). Right, the mean numbers of clips labelled 'pro' and 'con' chosen (642 observations from 321 independent participants). **b**, Fraction of participants willing to eat the insects for the promised incentive, averaged across species, by incentive and video condition (3,307 observations from 671 independent participants).

**c**, Cumulative distribution functions of ex-post surplus by incentive, using data from the Video condition. The data are left-censored because reservation prices are right-censored at US$60. **d**, Replication of **c** with the distribution of reservation prices imputed from the No-Video treatment with the US$3 incentive. The error bars in **a**,**b** display the 95% confidence intervals. Standard errors are clustered by participant.

video and then chose the video that ex-post rationalized this decision. To disprove this hypothesis, participants in a control condition completed the experiment without the possibility of selecting or watching any video or clips. I found that the information did affect subsequent choices. Raising the incentive caused participation to increase by 33 percentage points (from 38% to 71%; $t$-test, $t = 8.14$, $P = 0.000$) when participants could select a video but by only 23 percentage points (from 36% to 59%; $t$-test, $t = 4.51$, $P = 0.000$) when they could not, though the difference in differences did not reach statistical significance ($t$-test, $t = 1.62$, $P = 0.105$).

**Welfare.** I next examined how incentives affected welfare, measured as ex-post surplus. The interpretation of ex-post surplus as welfare assumes that handing out the insects conveyed substantial information. Indeed, the hand-out caused reservation prices to rise by US$6.68 ($t$-test, $t = 12.69$, $P < 0.000$) for some species and fall by US$1.77 ($t$-test, $t = −3.41$, $P = 0.001$) for others, compared with reservation prices elicited before the hand-out (Supplementary Information section A.2). The mean absolute change was US$6.68 ($t$-test, $t = 20.94$, $P = 0.000$). In the US$3 condition, around one fifth of all choices yielded a negative surplus (Fig. 2c). This number dropped to just under 10% in the US$30 condition. Moreover, raising the incentive did not increase the magnitude of negative surpluses, yet it greatly increased positive surpluses. Thus, there was nearly a stochastic dominance relationship between the surplus distributions across the incentive amounts. Almost any

social welfare function increasing in ex-post surplus will therefore favour the high incentive. No such function will support allowing the transaction at the low incentive but preventing it at the high incentive, contrary to UIH-normative (though planners that heavily weigh ex-post losses prefer preventing the transaction at all incentives).

There are two potential concerns about this analysis. First, the incentive might have directly affected ex-post reservation prices (for instance, through anchoring)[35]. Second, the videos themselves may have distorted reservation prices (for example, because they are tendentious). I simultaneously addressed these concerns by assuming that the true reservation prices in each treatment follow the distribution revealed by participants who could not watch a video and were given the US$3 incentive. I assigned the participant with the highest reservation price in the US$30, Video treatment the highest reservation price observed in the US$3, No-Video treatment. I performed a similar match for all other reservation price ranks. On the basis of these counterfactual reservation prices but using the participants' actual participation decisions, I calculated the counterfactual surplus. This measure incorporates the effects of incentives and information on participation decisions but excludes such effects on reservation prices. Figure 2d displays the resulting surplus distribution. Raising the incentive to US$30 increased not only welfare gains but also losses.

The higher incentive was therefore no longer unambiguously preferable. Yet, the increase in losses is not per se sufficient for UIH-normative. The reason is that UIH-normative requires two

**Table 1 | Tests for UIH-normative in Experiment 1**

|  | (1) | (2) | (3) | (4) |
|---|---|---|---|---|
| **Alternative welfare benchmark** |  | ✓ |  | ✓ |
| **Noise correction** |  |  | ✓ | ✓ |
| **Weight on losses** |  |  |  |  |
| Minimum for US$30 unacceptable, $\underline{\alpha}$ | 0.915 (0.016) | 0.801 (0.034) | 0.677 (0.036) | 0.618 (0.038) |
| Maximum for US$3 acceptable, $\overline{\alpha}$ | 0.183 (0.046) | 0 (0) | 0.134 (0.035) | 0 (0) |
| UIH-normative satisfied | No | No | No | No |
| $P$ value for $\underline{\alpha} = \overline{\alpha}$ | 0 | 0 | 0 | 0 |
| Participants | 400 | 400 | 400 | 400 |
| Observations | 1,921 | 1,921 | 1,921 | 1,921 |

UIH-normative requires $\underline{\alpha} \leq \overline{\alpha}$. Columns labelled 'Noise correction' predict the participants' reservation price for a species of insect using their reservation prices for the remaining four species in a ridge regression. Columns labelled 'Alternative welfare benchmark' use reservation prices imputed from the No-Video condition at the US$3 incentive. Standard errors are shown in parentheses, clustered by participant. $P$ values concern two-sided $z$-tests of the null hypothesis that $\underline{\alpha} = \overline{\alpha}$.

conditions to hold simultaneously. First, the weight on losses must be sufficiently high to render preventing the transaction at the high incentive optimal (total welfare at the high incentive is lower than that from preventing the transaction). Second, the weight on losses must be sufficiently low to make participating at the low incentive acceptable (total welfare at the low incentive exceeds that from preventing the transaction). Among the class of welfare functions that place weight $\alpha$ on losses and weight $(1 - \alpha)$ on gains, the first condition requires $\alpha \geq \underline{\alpha} = g(30)/(g(30) - l(30))$ and the second requires $\alpha \leq \overline{\alpha} = g(3)/(g(3) - l(3))$, where $g(m)$ and $l(m)$ denote total surplus gains and losses at incentive $m$ for all participants who participate in the transaction (Methods). The specific values of these bounds depend on the empirical distribution of gains and losses. Critically, it is possible that the lower bound $\underline{\alpha}$ exceeds the upper bound $\overline{\alpha}$. If so, it is not optimal to allow the transaction at capped incentives, no matter the weight on those who lose from the transaction.

Table 1 shows the estimates of the bounds $\underline{\alpha}$ and $\overline{\alpha}$ and the difference between them, depending on whether I use imputed reservation prices (columns 2 and 4) and on whether I account for the fact that reservation prices might have been elicited with noise, which would cause an incorrect assignment to gains and losses for some observations (columns 3 and 4; Methods). In each case, the difference between $\underline{\alpha}$ and $\overline{\alpha}$ is large and statistically highly significant, and precludes the existence of a weight on losses consistent with UIH-normative. These results apply separately for each insect species (Supplementary Information section A.3). UIH-normative was violated throughout.

### Experiment 2: choosing between advisors

The previous experiment leaves open two questions. First, will the effect of participation incentives differ if we measure decision quality without the benefit of hindsight but on the basis of the participant's information at the time of decision? Second, are there cases in which UIH-normative is more likely satisfied, such as when incentives for a transaction are paid immediately, but potential downsides only realize with a delay?

**Design.** Experiment 2 consisted of two stages that each proceeded in multiple rounds of which a random one was paid out. Fifty-eight German student participants completed this experiment, in addition to 348 who completed one of the extensions.

In each round of the first stage, a participant, endowed with €110, decided whether to risk losing €100 with a 50% chance.

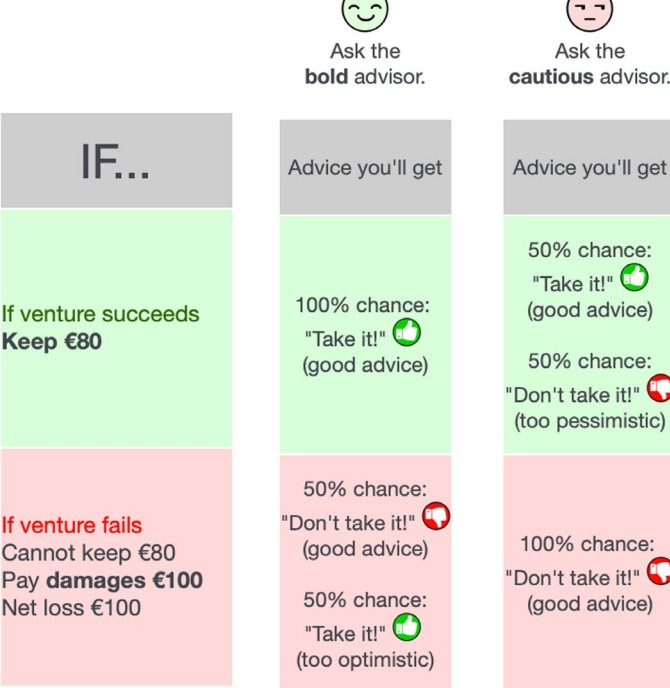

**Fig. 3 | Information choice in Experiment 2.** Screenshot of the participants' interface for the case $m = 80$, which is forfeited if the loss realizes.

A participant who agreed to take the risk received an incentive of €$m$, where $m \in \{20, 30, 70, 80\}$ varied across treatments. I varied whether the participant could or could not keep that payment if the loss materialized.

Crucially, before the participants decided whether to take the risk, but after they learned the amount $m$ they would receive in exchange for participation, they chose between two computerized advisors (Fig. 3). The advisors had imperfect foresight about whether the loss would occur. The 'Bold Advisor' was likely to recommend taking the risk. He always recommended it if no loss would occur. With a 50% chance, he also—erroneously—recommended taking the risk when it would cause a loss. The 'Cautious Advisor' was likely to recommend rejecting the gamble. He always recommended rejection if the loss would materialize. With a 50% chance, he also—erroneously—recommended rejection if no loss would occur. UIH-positive predicts that participants will more often opt for the Bold Advisor if $m$ is higher. The participants made 18 decisions such as this, with one decision from the entire experiment randomly selected to determine the participants' study payment (Methods).

According to UIH-normative, a higher incentive causes worse decision-making. It may happen, for instance, that an €80 incentive causes participants to choose the Bold Advisor and take his recommendation at face value, unaware that his positive recommendation implies merely a 2/3 success probability. (By Bayes' law, $P$(success|positive recommendation from Bold Advisor) = $\frac{0.5 \times 1}{0.5 \times 1 + 0.5 \times 0.5} = \frac{2}{3}$.) If so, they will participate too often, against their own interest, if they are risk-averse.

I measured the presence and severity of such mistakes according to the participants' own preferences. To this end, the second stage of the experiment presented the participants with a series of lotteries. The participants revealed the certain amount of money they considered just as good as the lottery (the certainty equivalent), elicited in a way that ensured truth-telling (Methods), for each lottery. Unbeknownst to the participants, each lottery corresponded to a decision from the first stage. For instance, consider a participant who, in the first stage, faced

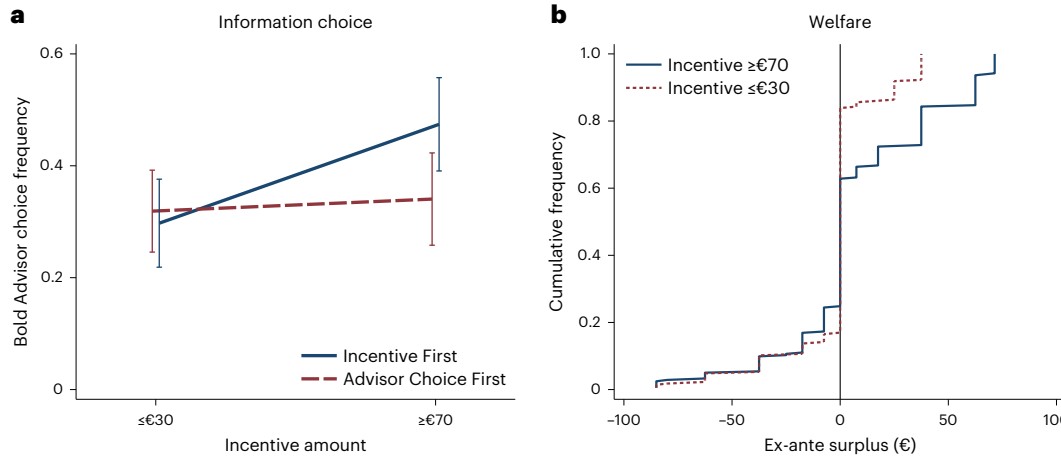

**Fig. 4 | Results from Experiment 2. a**, Fraction of participants choosing the Bold Advisor depending on whether the incentive amount was known when choosing (928 observations from 58 independent participants). The error bars display the 95% confidence intervals. Standard errors are clustered by participant. **b**, Cumulative distribution functions of ex-ante surplus by incentive.

incentive $m = 80$, which she would lose if the risk materialized, and who observed a positive recommendation from the Bold Advisor. The corresponding stage-2 decision was a win €80/lose €100 gamble with a 2/3 success probability, described as such. If the participant accepted the risk in stage 1 but revealed a negative certainty equivalent in stage 2, her stage-1 decision was mistaken, and the magnitude of the certainty equivalent was its cost to the participant. When it was positive, the certainty equivalent showed how much the participant profited from the opportunity to participate in the transaction at the given incentive.

As supporting evidence, each round of the first stage also elicited subjective beliefs about whether the loss would materialize, after the participant had seen the advisor's recommendation. Rational behaviour requires that mean beliefs do not depend on the incentive amount (law of iterated expectations).

**Information choice.** Consistent with UIH-positive, an increase in the incentive from €30 or below to €70 or above increased the choice of the Bold Advisor from 29.7% to 47.4% ($t$-test, $t = 3.84$, $P = 0.000$; Fig. 4a). This effect substantially exceeds that of a condition in which participants chose an advisor before knowing the incentive they would face ($t$-test, $t = 2.98$, $P = 0.004$). The change in advisor preference also affected participants' beliefs that the gamble would pay off. If the incentive amount was €30 or lower, the mean subjective posteriors were 48.9%, which is statistically indistinguishable from the Bayesian expected posterior of 50% ($t$-test, $t = -0.49$, $P = 0.627$). If the incentive amount increased to €70 or more, the mean subjective posteriors increased by 7.7 percentage points ($t$-test, $t = 2.39$, $P = 0.020$), to 56.6% (exceeds 50%; $t$-test, $t = 3.06$, $P = 0.003$). This effect is inconsistent with rational inference and suggests potentially harmful effects of higher incentives on welfare.

**Welfare.** Figure 4b shows the effect of incentives on welfare. For incentives €30 or lower, just under 20% of decisions were mistaken. For incentives €70 or more, the fraction was slightly higher. Yet, the magnitude of positive certainty equivalents increased greatly.

Because raising the incentive was not unambiguously good, I tested UIH-normative formally. As in Experiment 1, I estimated the maximal weight one may place on losses while still deeming the low incentive acceptable as $\overline{\alpha} = \frac{g(20)}{g(20)+l(20)}$ and the minimal weight one must place on losses to deem the high incentive unacceptable as $\underline{\alpha} = \frac{g(80)}{g(80)+l(80)}$. Here, the participants' ex-ante gain $g$ or loss $l$ from a transaction they accepted in stage 1 equals their certainty equivalent

for the corresponding lottery elicited in stage 2. If they rejected the transaction, the value is zero. I used seemingly unrelated regression to estimate mean gains and losses at €20 and €80 using all four incentive amounts (Methods).

I found that regardless of whether the incentive payment was lost if the risk materialized, the lower bound $\underline{\alpha}$ exceeded the upper bound $\overline{\alpha}$ by more than 50% (Table 2). Hence, regardless of the weight on losses from the transaction, it was never optimal to permit the transaction at €20 but not at €80. The threshold for preventing the transaction altogether was lower when the risk entailed losing the incentive payment than when it did not, but UIH-normative remained violated in either case.

These results continued to hold when I replaced the ex-ante welfare benchmark with a benchmark that measures whether decisions were good in hindsight, and when I controlled for noise in certainty equivalents (Supplementary Information section B).

I replicated these results in an experiment with a lower overall stake size with 348 participants (see 'Experiment 2' in Methods). I included treatments to address the following two concerns. First, incentives may cause bad decision-making when the risk materializes much later than the incentive payment. In the corresponding treatment, the participants received incentive payments immediately, but potential losses took effect only with a three-month delay (discounted from a fixed payment they would otherwise receive at that date). Second, I varied the prior risk of loss between 20% and 80%. This treatment might raise error rates—for example, for individuals subject to base rate neglect[36]. If such difficulties interact with the incentive amount, UIH-normative might be satisfied.

In these extensions, UIH-normative was robustly violated (Supplementary Information section B). While some treatments increased choice mistakes, these changes did not strongly interact with the incentive amount. Such cases present a stronger argument for preventing the transaction altogether, but they do not support capping incentives.

A separate experiment (Supplementary Information section C) tested whether UIH-normative is more likely to be satisfied if participants search through a large amount of information rather than observing a single recommendation from an advisor. There, too, choices satisfied UIH-positive but violated UIH-normative. In fact, higher incentives increased efforts to determine whether participation was the right choice.

### Effect of incentives on perfectly rational agents

To understand the simultaneous confirmation of UIH-positive and refutation of UIH-normative in the preceding experiments, consider

**Table 2 | Tests for UIH-normative in Experiment 2**

| Potential downside | Treatment | |
|---|---|---|
| | €100 | €100−incentive |
| **Weight on losses** | | |
| Minimum for US$30 unacceptable, $\underline{\alpha}$ | 0.605 (0.068) | 0.818 (0.066) |
| Maximum for US$3 acceptable, $\overline{\alpha}$ | 0.386 (0.106) | 0.451 (0.120) |
| *P* value for $\underline{\alpha} = \overline{\alpha}$ | 0.084 | 0.004 |
| UIH-normative satisfied | No | No |
| Participants | 58 | 58 |
| Observations | 232 | 232 |

UIH-normative requires $\underline{\alpha} \leq \overline{\alpha}$. Standard errors are shown in parentheses, clustered by participant. *P* values concern two-sided *z*-tests of the null hypothesis that $\overline{\alpha} = \underline{\alpha}$. ***$P < 0.01$.

an entirely rational decision maker who decides what information to consult before choosing whether to participate (the formal model is in Supplementary Information section D). Because rational agents do not make mistakes, UIH-normative is violated by construction. Yet, this agent's information choice behaviour will conform to UIH-positive.

The reason is intuitive. Decision makers who are constrained in how much information they can consult can make two types of mistakes. First, they may participate, even though they would have abstained under complete information. Second, they may abstain, even though they would have participated under complete information. Rational decision makers will select information sources that help them make more expensive mistakes less often. Incentives for participation change the costs of the two mistakes. With a low incentive, decision makers have little to gain from participating but potentially much to lose. Therefore, they will rationally target their information search towards sources that help prevent mistaken participation, even if this comes at the cost of abstaining by mistake more often (the discouraging video and the Cautious Advisor in Experiments 1 and 2, respectively). By contrast, if the incentive is high, they will target their search towards sources more likely to prevent mistaken abstention, even if that comes at the cost of participating by mistake more often (the encouraging video and the Bold Advisor). Mechanically, these information sources will more often recommend participation than those that decision makers seek out under low incentives. The behaviour of rational decision makers is therefore consistent with UIH-positive.

## Discussion

Overall, this paper shows that incentives exert the behavioural effects that the undue inducement literature hypothesizes. Yet, from a consequentialist point of view, these effects do not justify that literature's normative conclusions. A formal model of costly information acquisition provides a unifying explanation.

Given the potentially high costs of preventing voluntary transactions, experiments paralleling those reported here should be conducted in the field. Unless their results differ drastically from the current ones, the rules and guidelines restricting incentives due to undue inducement concerns should be reconsidered.

It is important to distinguish my results from two related intuitions. First, some authors worry that incentives would disproportionately attract the poor. Concerns about inequality are logically distinct from concerns about undue inducement. The latter deems high incentives problematic per se; it applies even to a hypothetical world without any inequality[37]. I have theoretically and experimentally examined the relation between inequality and incentives elsewhere[38]. A second concern is the (empirically partly disproved[39–42]) hypothesis that incentives decrease the supply of willing participants by lowering altruistic participation[43,44]. Concerns about supplied quantities are

distinct from concerns about the welfare of the incentivized. The UIH addresses only the latter.

Future research should test the robustness of my findings in other contexts, with different participant pools and with transactions currently subject to undue-inducement-based regulation, such as female egg donation or gestational surrogacy. These cases might recruit psychological mechanisms not present in the current study, such as choking under pressure[45]. To address non-consequentialist concepts such as autonomy and consent, future research should also extend the empirical study of undue inducement beyond the welfarist framework used here.

## Methods
### Experiment 1
This research complies with all relevant ethical regulations. It was approved by Stanford University's Non-medical Institutional Review Board (IRB). Informed consent was obtained from each participant. All participants earned a completion payment of US$35, except five participants who reneged on their decision to eat insects. Those participants' completion payments were reduced to US$15. In addition, each participant received payment in exchange for ingesting insects if they had decided to do so and followed through with that decision. Of the total of 671 participants, 313 were women, 353 were men and 5 were non-binary. The mean age was 21.56 years.

**Additional design information.** To increase statistical power, each participant made choices about five different 'food items' in each stage of the experiment: two house crickets, five large mealworms, three silkworm pupae, two mole crickets and two field crickets (Fig. 1). Even in countries such as China and Mexico, many people do not practise insect eating. Instead, it is concentrated within particular regions and communities. In my data, Asians and Hispanics were just as unwilling to eat insects as white Americans.

The participants were not allowed to watch both videos. This design choice increases the chance of finding harmful effects of incentives. It models the fact that learning about transactions subject to undue inducement concerns typically costs substantial effort and time, which constrains the amount of information that can be processed.

For each item, the participants revealed their reservation price on a single page with 21 binary choices between the alternatives 'Get $p. In exchange, eat the food item' and 'Do not participate in this transaction', where $p \in \{1, 2, 3, 4, 6, 8, 10, 12.5, 15, 17.5, 20, 22.5, 25, 27.5, 30, 33, 36, 39, 44, 50, 60\}$. (The amount US$3 was not included in the decision lists for the first 79 participants.) The participants clicked on the line at which they preferred to switch from refusing the transaction to accepting it. The remaining choices were filled in automatically.

To force the participants to view, touch and smell each insect, within each container the insects sat atop a folded strip of paper with a code. The participants had to remove each strip and enter the code into the computer. The participants completed filler tasks during the handing out of the insects (an extended Cognitive Response Scale[46] and sets D and E of Raven's Standard Progressive Matrices[47]).

**Incentives.** Each participant made multiple decisions. To incentivize truthful preference revelation, the participants learned that exactly one of all decisions would be randomly chosen for implementation at the end of the experiment. That decision would entirely determine a participant's payment and consumption of insects.

The participants knew that their decision whether to eat insects in exchange for the initially promised amount would determine the outcome with 80% probability. The first and second reservation price elicitation each determined the participants' outcome with a 7% chance. With the remaining 6% chance, the participants' outcome was determined according to a final stage that came as a surprise to each participant in which they predicted other participants' reservation prices.

To minimize social motives such as trying to impress other participants, the participants knew that all insects would be consumed in a visually secluded space in the presence of only the experimenter, who would ensure that the participant consumed the animals in their entirety.

The participants made some consumption decisions before seeing the actual insects and thus could be unpleasantly surprised. To ensure that the participants did not accept transactions expecting to renege on their choice, participants who did not honour their commitment to eat the insects in exchange for money forewent the promised amount and lost an additional US$20, discounted from a US$35 completion payment.

**Data collection.** Each of the 671 participants participated in one of 39 computerized sessions run on Qualtrics lasting about 2.5 hours each in early summer 2015 at the Ohio State University (499 participants), Stanford University (110 participants) and the University of Michigan (62 participants), recruited using the laboratories' experimental economics participant databases. A total of 271 participants participated in the No-Video condition (136 with the US$3 incentive and 135 with the US$30 incentive), and 400 participated in the Video condition (197 with the US$3 incentive and 203 with the US$30 incentive). Each session involved both payment conditions, but either all or none of the participants in a session were in the Video condition and did not make decisions about field crickets. Participants at Stanford in the Video condition (79 participants) did not select video clips. All participants knew that the Stanford IRB had approved the experiment and that all insects were produced for human consumption.

Five participants (0.8%) refused to follow through when one of their decisions to eat an insect was selected for implementation, and they paid the US$20 penalty. Of these participants, all were in the US$30 condition, four were in the Video condition and three had seen the encouraging video. Randomization into treatments was successful. Of 24 $F$-tests for differences in participants' predetermined characteristics across the four treatments, only 1 was significant at the 5% level (Supplementary Information section A.1).

**Analysis.** The calculation of $\underline{\alpha}$ and $\bar{\alpha}$ is based on the following Bergson–Samuelson social welfare function: $W_\alpha = \sum_{i=1}^n f_\alpha(w_i)$ with $f_\alpha(w) = \alpha$ if $w < 0$ and $f_\alpha(w) = (1 - \alpha)$ if $w \geq 0$. This function is utilitarian if $\alpha = \frac{1}{2}$. The pair consisting of mean gain $g(m) = \frac{1}{n} \sum_{i=1}^n \max(w_i(m), 0)$ and mean loss $l(m) = \frac{1}{n} \sum_{i=1}^n \min(w_i(m), 0)$ is a sufficient statistic for this class of welfare functions. The transaction is acceptable at incentive $m$ if welfare from permitting it, $(1 - \alpha)g(m) + \alpha l(m)$, exceeds welfare from preventing it, which is zero. Rearranging this inequality at $m = 3$ yields $\alpha \leq \frac{g(3)}{g(3) - l(3)} = \underline{\alpha}$. Rearranging the reverse inequality at $m = 30$ yields $\alpha \geq \frac{g(30)}{g(30) - l(30)} = \bar{\alpha}$, as used in the main text.

Elicited reservation prices are right-censored at US$60. To calculate $\underline{\alpha}$ and $\bar{\alpha}$ in Table 1, I accounted for the censoring by fitting a log-normal distribution to the empirical distribution of reservation prices that exceed the median. I then replaced the censored observations with the implied expected value. The predicted mean reservation prices for censored observations were US$107 in the low-incentive condition and US$126 in the high-incentive condition.

To average out elicitation noise in columns 3 and 4 of Table 1, I predicted each participant's reservation price for species $r$ using the participant's choices for all remaining species. The predictive model is ridge-regression estimated on the full sample with a penalty term selected using tenfold cross-validation[48]. I recalculated the implied bounds $\underline{\alpha}$ and $\bar{\alpha}$ on the basis of these predictions.

## Experiment 2

This research complies with all relevant ethical regulations. It was approved by the IRB of the Economics Department of the University of Zurich. Informed consent was obtained on screen.

Of the 58 participants who completed this experiment, 38 were women and 20 were men, with a mean age of 27.05 years. Of the 348 participants who completed the Extension Experiment, 208 were women, and 140 were men, with a mean age of 26.63 years.

**Study structure.** The treatments varied along three dimensions, administered within-participant. The first dimension was incentive amount, $m \in \{20, 30, 70, 80\}$. The second was information choice. In the Incentive First condition, the participants learned the incentive before selecting an advisor. The Advisor Choice First condition asked participants to choose an advisor before learning the incentive amount, so they could not tailor information choice to the specific incentive they would face. The third dimension was downside. In the Limited Downside condition, participants received the participation payment $m$ even if the loss materialized. In the Large Downside condition, participants received the participation payment $m$ only if the loss did not materialize.

In the first stage of the study, the participants completed 18 rounds in random order. Sixteen rounds presented them with all combinations of (1) the four incentive amounts, (2) the Incentive First and Advisor Choice First conditions, and (3) the Limited and Large Downside conditions. The two remaining rounds served as an attention check. They offered incentive $m = 0$ in the Incentive First condition. The participants did not learn the realization of any gamble in any round.

An elicitation of the participants' posterior belief followed each round. The participants selected one of 12 bins, corresponding to 0%, 1–10%, 10–19%, 20–29%, …, 80–89%, 90–99% or 100% certainty that the state in that round was good. The participants could return from the belief elicitation stage to the betting stage to change their participation decision.

In the second stage of the study, the participants faced 26 multiple decision lists consisting of 11 questions of the form 'Which of the following two options do you prefer?' Alternative 1 was 'Win €$m$ with $\gamma \times 100\%$ chance, lose $L$ with $(1 - \gamma) \times 100\%$ chance'; alternative 2 was 'Receive €$c$ for sure' (if $c \geq 0$) or 'Lose €$c$ for sure' (if $c < 0$). The parameters $m$, $L$ and $\gamma$ were fixed in each round, while $c \in \{-85, -75, -50, -25, -15, 0, 15, 25, 50, 75, 85\}$ across the 11 questions, in random order.

Each round corresponds to a different vector $(m, L, \gamma)$. For each of the four levels of incentive amounts $m$ and for each of the Limited and Large Downside conditions, the participant completed one decision list in which the success probability $\gamma$ equalled the Bayesian posterior from a recommendation to participate by the Bold Advisor (2/3) and another list in which it equalled the Bayesian posterior from a recommendation to abstain by the Cautious Advisor (1/3). The participants also completed decision lists for the posteriors corresponding to a recommendation to abstain by the Bold Advisor (which is zero) and corresponding to a recommendation to participate by the Cautious Advisor (which is one). For each of these degenerate posteriors, the participants completed four lists corresponding to $m = 30$ and $m = 70$ for each of the Limited and Large Downside conditions.

**Randomization.** Within each stage of the study, the rounds were randomized at the individual level. The advisors were presented next to each other. The Bold Advisor was displayed on the left for a random half of participants and on the right for the remaining half. The decisions in each list in stage 2 appeared in random order, drawn anew in each round.

**Framing.** In stage 1 of the experiment, the participants decided whether to accept a 'venture' that could either 'succeed' or 'fail', in exchange for a 'venture participation payment'. If the venture succeeded, they could keep the venture participation payment, and no further consequences occurred. If the venture failed, they had to 'pay damages'. Advisor recommendations read either 'The [type] advisor recommends: Participate in the venture!' accompanied by a thumbs-up symbol on a green

background, or 'The [type] advisor recommends: Don't participate in the venture!' accompanied by a thumbs-down symbol on a red background. Each survey round alerted the participant that a new state was drawn using a short animation that required the participant to click to shuffle and randomly pick an unobserved outcome.

**Incentives.** The study payment was entirely determined by a single, randomly drawn decision, which incentivized truthful revelation of preferences. With a 2/3 chance, that decision was from stage 1. With the remaining 1/3 chance, it was from stage 2. In the case of the former, the payment was determined with an 80% chance by the decision whether to accept the transaction and whether the loss materialized. With the remaining 20% chance, it was determined by the belief elicitation, which was incentivized by the probabilistic quadratic scoring rule[49]. Incentive compatibility requires only that the participants knew that any single decision could determine their payment, not that the participants remembered the implementation probabilities.

**Extensions.** Using smaller overall stake sizes, I tested whether the timing of consequences and the prior success probability affect conclusions about UIH-normative. The participants received a completion payment of €15. They risked losing €6 by accepting a transaction. They faced incentive amounts $m \in \{1, 2, 4, 5\}$. The second stage used certain amounts $c \in \{-6, -4.5, -3, -1.5, -0.5, 0, 0.5, 1.5, 3, 4.5, 6\}$ across 11 questions on each page. For context, the purchasing power parity exchange rate in 2020 was €1 = US$1.38. The University of Cologne estimates monthly student living expenses of €832.

To maintain comparability to the main experiment, I replicated that experiment with the lower stake size. In addition to the three within-participant treatments, I administered two treatments across participants. The first was prior risk probabilities, $\mu \in \{0.2, 0.5, 0.8\}$. The risk remained constant throughout the study for any given participant. The second treatment was the delay condition. Participants in that condition received incentive payments immediately, but potential losses took effect only with a three-month delay. Delayed losses were discounted from a €6 payment that the participants otherwise received with a three-month delay. The participation payment immediately disbursed to these participants was €9, thus bringing the total unconditional payment to €15, as was the case for participants not in the Delayed Consequences condition. In stage 2 of the study, certain payments were presented as 'Your immediate payment rises [falls] by €c for sure.' In the Large Downside condition, the lottery read 'With $\gamma \times 100\%$ chance: your immediate payment rises by €m, and your delayed payment stays unchanged. With $(1 - \gamma) \times 100\%$ chance: your immediate payment stays unchanged, and you lose €L of your delayed payment.' In the Limited Downside condition, the text 'your immediate payment stays unchanged' was replaced by 'your immediate payment rises by €m.'

**Data collection.** I conducted the study online with the Qualtrics research survey tool using the participant pool and procedures of the Cologne Laboratory for Economic Research at the University of Cologne, disbursing payment through PayPal. A total of 58 unique participants participated on 7 February 2022. In addition, 348 unique participants completed one of the extension treatments on 25–27 November 2020. In the Contemporaneous Consequences condition, 53, 64 and 58 participants participated with a prior success probability of 20%, 50% and 80%, respectively. The corresponding participant counts for the Delayed Consequences condition are 64, 50 and 59. The participants took between 45 and 90 minutes to complete the study.

All participants had to pass two comprehension checks before making decisions. The first check required the participants to correctly mark each of nine statements as true or false. The second check consisted of six such statements. Participants who answered incorrectly did not receive feedback on which statement was mismarked. Hence,

participants were highly unlikely to pass by chance or trial and error. The participants were asked to review the instructions until they passed the comprehension checks.

**Attrition.** There was no attrition in the main experiment. In the Extension Experiment, the 348 complete responses made up 95.6% of the 364 surveys started on a machine satisfying the technical requirements (desktop or laptop computers with sufficient screen width). Attrition was unrelated to treatment. Importantly, the crucial variation of incentive amounts and information order occured within participants.

**Certainty equivalents.** Some participants chose a certain amount $c$ over a given lottery but chose that lottery over some larger sure amount $c'$. Averaging across all participants, such non-monotonicities occurred in 2.3 of the 26 rounds (median, 2). The corresponding number in the Extension Experiment was 2.9 (median, 2). In case of a non-monotonicity, I defined the participant's certainty equivalent as the lowest certain amount the participant preferred to the lottery. (The results were qualitatively unchanged if I included only observations that respect monotonicity.)

**Estimation of UIH-normative.** To use all four incentive amounts in the formal test of UIH-normative, I estimated the following linear system using seemingly unrelated regression:

$$g_i(m) = \beta_0^G + \beta_1^G m + \epsilon_i^G(m)$$
$$l_i(m) = \beta_0^L + \beta_1^L m + \epsilon_i^L(m). \tag{1}$$

Here, $\epsilon_i^G(m)$ and $\epsilon_i^L(m)$ are independent individual effects with expectation zero such that $g_i(m) \geq 0 \geq l_i(m)$ for all $i$. I imputed $g(m) = \hat{\beta}_0^G + \hat{\beta}_1^G m$ and $l(m) = \hat{\beta}_0^L + \hat{\beta}_1^L m$, where $\hat{\beta}_i^j$ indicates a parameter estimate, to calculate the bounds $\underline{\alpha}$ and $\overline{\alpha}$ using $m = 20$ and $m = 80$ for the low and high incentive amounts, respectively.

### Reporting summary

Further information on research design is available in the Nature Portfolio Reporting Summary linked to this article.

## Data availability

The datasets generated and analysed during the current study are available in a replication package on the Harvard Dataverse at https://doi.org/10.7910/DVN/3PFZKP. Source data are provided with this paper.

## Code availability

The replication package includes all Stata code to replicate the statistical analysis. It is available at https://doi.org/10.7910/DVN/3PFZKP.

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

## Acknowledgements

I thank my PhD advisors M. Niederle, B. D. Bernheim and A. E. Roth. I thank Y. Chen, P. J. Healy and A. Ockenfels for providing experimental laboratory resources and E. Küpper for excellent research assistance. Experiment 1 was funded by the Department of Economics at Stanford University. Experiment 2 was funded by the Department of Economics at the University of Zurich. Moreover, this project has received funding from the European Research Council under the European Union's Horizon 2020 research and innovation programme (grant agreement no. 741409). The paper reflects the author's view; the European

Research Council is not responsible for any use that may be made of the information it contains. The funders had no role in study design, data collection and analysis, decision to publish or preparation of the manuscript.

## Funding

## Competing interests

The author declares no competing interests.

## Additional information

**Correspondence and requests for materials** should be addressed to Sandro Ambuehl.

# Reporting Summary

## Statistics

For all statistical analyses, confirm that the following items are present in the figure legend, table legend, main text, or Methods section.

| n/a | Confirmed | |
|---|---|---|
| ☐ | ☒ | The exact sample size (*n*) for each experimental group/condition, given as a discrete number and unit of measurement |
| ☐ | ☒ | A statement on whether measurements were taken from distinct samples or whether the same sample was measured repeatedly |
| ☐ | ☒ | The statistical test(s) used AND whether they are one- or two-sided *Only common tests should be described solely by name; describe more complex techniques in the Methods section.* |
| ☐ | ☒ | A description of all covariates tested |
| ☐ | ☒ | A description of any assumptions or corrections, such as tests of normality and adjustment for multiple comparisons |
| ☐ | ☒ | A full description of the statistical parameters including central tendency (e.g. means) or other basic estimates (e.g. regression coefficient) AND variation (e.g. standard deviation) or associated estimates of uncertainty (e.g. confidence intervals) |
| ☐ | ☒ | For null hypothesis testing, the test statistic (e.g. $F$, $t$, $r$) with confidence intervals, effect sizes, degrees of freedom and $P$ value noted *Give P values as exact values whenever suitable.* |
| ☒ | ☐ | For Bayesian analysis, information on the choice of priors and Markov chain Monte Carlo settings |
| ☒ | ☐ | For hierarchical and complex designs, identification of the appropriate level for tests and full reporting of outcomes |
| ☒ | ☐ | Estimates of effect sizes (e.g. Cohen's *d*, Pearson's *r*), indicating how they were calculated |

*Our web collection on statistics for biologists contains articles on many of the points above.*

## Software and code

Policy information about availability of computer code

| Data collection | All data in this study were collected with the Qualtrics Research Core survey tool. |
|---|---|
| Data analysis | All analysis in this study was done using Stata/MP 16.0 for Mac (64-bit Intel) |

For manuscripts utilizing custom algorithms or software that are central to the research but not yet described in published literature, software must be made available to editors and reviewers. We strongly encourage code deposition in a community repository (e.g. GitHub). See the Nature Portfolio guidelines for submitting code & software for further information.

## Data

Policy information about availability of data

All manuscripts must include a data availability statement. This statement should provide the following information, where applicable:
- Accession codes, unique identifiers, or web links for publicly available datasets
- A description of any restrictions on data availability
- For clinical datasets or third party data, please ensure that the statement adheres to our policy

The datasets generated during and analyzed during the current study are available in a replication package on the Harvard Dataverse, https://doi.org/10.7910/DVN/3PFZKP.

# Human research participants

Policy information about <u>studies involving human research participants and Sex and Gender in Research.</u>

| | |
|---|---|
| Reporting on sex and gender | All analyses pool across sexes / genders. The terms "sex" and "gender" do not appear in the manuscript. Appendix table B.1 lists the fraction of (self-reported) male participants across treatments in Experiment 1. Following established standards in experimental economics, I do not perform gender-based analysis. Based on concerns about forking paths and fishing for statistical significance, these standards advise researchers not to perform such subgroup analyses unless the are a main focus of the study design which is not the case in the present study. |
| Population characteristics | Participants in Experiment 1 were undergraduate students at Stanford University, the Ohio State University, and the University of Michigan. Of the total of 671 subjects, 313 were women, 353 were men, and 5 were nonbinary. Mean age was 21.56 years. Participants in Experiment 2 and the Extension Experiment were undergraduate students at the University of Cologne. Of the 58 subjects who completed this experiment, 38 were women and 20 were men, with a mean age of 27.05. Of the 348 subjects who completed the Extension Experiment, 208 were women, and 140 were men, with a mean age of 26.63. |
| Recruitment | Recruitment was done by sending invitation emails to the subject pools of the experimental economics laboratories at each university where a session was run. Selection into the study was limited by the fact that invitation emails did not reveal any study details. Assignment to treatment was randomized across (experiments 1 and 2) as well as within (experiment 2) subject. Treatment effect estimates are unbiased for the study population due to random assignment to treatment. |
| Ethics oversight | Experiment was approved by Stanford University's Non-medical IRB in protocol #34001. Experiment 2 was approved by the University of Zurich in protocol OEC IRB # 2022-007. |

Note that full information on the approval of the study protocol must also be provided in the manuscript.

# Field-specific reporting

Please select the one below that is the best fit for your research. If you are not sure, read the appropriate sections before making your selection.

☐ Life sciences    ☒ Behavioural & social sciences    ☐ Ecological, evolutionary & environmental sciences

For a reference copy of the document with all sections, see nature.com/documents/nr-reporting-summary-flat.pdf

# Behavioural & social sciences study design

All studies must disclose on these points even when the disclosure is negative.

| | |
|---|---|
| Study description | The experiments in this study follow the standards of experimental economics. Human subjects make decisions that have consequences for the payment they will receive for participation and for any other activities they will have to perform after the study. Measurement is quantitative. |
| Research sample | Experiment 1 uses US undergraduate students at three universities. The study sample was chosen following standards in experimental economics (which typically use such subject pools) as well as due to the need for physical interaction with participants. The sample is not representative of the general US population. Of the total of 671 subjects, 313 were women, 353 were men, and 5 were nonbinary. Mean age was 21.56 years. Experiment 2 and the Extension Experiment use German university students in Cologne who participated online. The study sample was chosen following standards in experimental economics during the Covid19-pandemic that precluded in-person studies. The sample is not representative of the general German population. Of the 58 subjects who completed this experiment, 38 were women and 20 were men, with a mean age of 27.05. Of the 348 subjects who completed the Extension Experiment, 208 were women, and 140 were men, with a mean age of 26.63. |
| Sampling strategy | Samples for both experiments are convenience samples. The laboratories at which the experiments were conducted maintain databases of subjects interested in participation who were invited by email. For experiment 1, Power simulations revealed that on the order of 800 subjects would be required, though due to unexpectedly low subject availability at the University of Michigan a sample of 671 participants was ultimately obtained. For experiment 2, sample size was determined based on the results of a pilot experiment conducted at the University of Toronto (which included only one of the two main outcome measures). |
| Data collection | All data in both experiments were collected with the Qualtrics Research Core survey tool. In experiment 1, only the researcher was present in the experimental sessions. Between 10 and 30 subjects participated in each session and made decisions on individual computer terminals. All subjects in a given session either participated in the video condition or in the no-video condition. Hence, the researcher was not blind to those treatments. Assignment to incentive condition was randomized by the computer on the subject level in each session, rendering the experimenter blind to this treatment assignment. In experiment 2, subjects participated online |

|  | and assignment to treatment was randomized by the computer, rendering the experimenter blind to treatment assignment. |
|---|---|
| Timing | Data for experiment 1 were collected in 39 sessions in May, June, and July 2015. Data for Experiment 2 were collected on February 7, 2022. Data for the Extension Experiment were collected November 25 to 27, 2020. |
| Data exclusions | No data were excluded from analysis. |
| Non-participation | No participants dropped out or declined participation in experiment 1. In Experiment 2, there was no attrition. In the Extension Experiment, the 348 complete responses make up 95.6\% of the 364 surveys that were started on a machine that satisfied the technical requirements (desktop or laptop computers with sufficient screen width). Out of the 16 attriters, 3 did not proceed to the first comprehension check, 10 dropped out at the first comprehension check, 1 dropped out at the second comprehension check, and 2 left the survey after the second comprehension check. Attrition is unrelated to treatment. The variation of incentive amounts and information order occur within subject. Hence, the corresponding main treatment estimates are unaffected by differential attrition. |
| Randomization | Partcipants were randomly assigned to treatment by the Qualtrics survey in both experiment 1 and experiment 2. Assignment to video condition in experiment 1 occurred at the session level. |

# Reporting for specific materials, systems and methods

We require information from authors about some types of materials, experimental systems and methods used in many studies. Here, indicate whether each material, system or method listed is relevant to your study. If you are not sure if a list item applies to your research, read the appropriate section before selecting a response.

## Materials & experimental systems

| n/a | Involved in the study |
|---|---|
| ☒ ☐ | Antibodies |
| ☒ ☐ | Eukaryotic cell lines |
| ☒ ☐ | Palaeontology and archaeology |
| ☒ ☐ | Animals and other organisms |
| ☒ ☐ | Clinical data |
| ☒ ☐ | Dual use research of concern |

## Methods

| n/a | Involved in the study |
|---|---|
| ☒ ☐ | ChIP-seq |
| ☒ ☐ | Flow cytometry |
| ☒ ☐ | MRI-based neuroimaging |

