## [Peer Review File · Nature Human Behaviour]

Peer Review Information

Journal: Nature Human Behaviour

Manuscript Title: An experimental test of whether financial incentives cause bad decisions

Corresponding author name(s): Sandro Ambuehl

Reviewer Comments & Decisions:

Decision Letter, initial version:

17th May 2023

Dear Professor Ambuehl,

Thank you once again for your manuscript, entitled "Can Incentives Cause Harm? Tests of Undue Inducement," and for your patience during the peer review process. I apologize again for the delay.

Your manuscript has now been evaluated by 4 reviewers, whose comments are included at the end of this letter. Although the reviewers find your work to be of interest, they also raise some important concerns. We are interested in the possibility of publishing your study in Nature Human Behaviour, but would like to consider your response to these concerns in the form of a revised manuscript that addresses all of the reviewers' points before we make a decision on publication.

Please note that regarding Reviewer #1's points 1 and 2, we agree with the reviewer that these are important points and they must be fully discussed as limitations to the study. However, we do not consider them to be a barrier to publication.

In sum, we invite you to revise your manuscript taking into account all reviewer and editor comments. We are committed to providing a fair and constructive peer-review process. Do not hesitate to contact us if there are specific requests from the reviewers that you believe are technically impossible or unlikely to yield a meaningful outcome.

We hope to receive your revised manuscript within two months. I would be grateful if you could contact us as soon as possible if you foresee difficulties with meeting this target resubmission date.

- Include a "Response to the editors and reviewers" document detailing, point-by-point, how you addressed each editor and referee comment. If no action was taken to address a point, you must provide a compelling argument. When formatting this document, please respond to each reviewer comment individually, including the full text of the reviewer comment verbatim followed by your response to the individual point. This response will be used by the editors to evaluate your revision and sent back to the reviewers along with the revised manuscript.
- Highlight all changes made to your manuscript or provide us with a version that tracks changes.

[REDACTED]

We look forward to seeing the revised manuscript and thank you for the opportunity to review your work. Please do not hesitate to contact me if you have any questions or would like to discuss these revisions further.

Sincerely,

Jamie

Dr Jamie Horder
Senior Editor
Nature Human Behaviour

REVIEWER COMMENTS:

Reviewer #1:

Remarks to the Author:

This paper uses behavioral empirical welfare methods to address the notion that undue influence can stem from outsized incentives in experiments. This is a considerably important issue particularly in considering RCTs designed for developing country contexts. Thus, the work is extremely policy and research relevant. That said, I have three major concerns with the manuscript as it is currently written.

1. My first concern is whether the experiments are able to address the topic of interest. This is a substantive concern. The authors use a decision to consume insects as a stand in for decisions that would hold substantive risk of harm. My guess is that the subject pool is at least somewhat aware of efforts to incorporate insects in western diets and would not perceive this is holding some long term risk of harm. Without that, the risk they are mitigating is perhaps just a risk of disgust--perhaps a much

smaller effect and one that may not be a good behavioral analog. In my read of the manuscript I felt like the authors glossed over this potential, which is honestly very key to whether the paper has merit. Is there some way to grapple with this?

2. The subject pool appears to be university students at highly ranked US schools. This is a very common criticism of experimental economics, but perhaps uniquely important in this endeavor. The scale of rewards (up to \$30) are not likely to be so outside the everyday experience of this pool as to be compelling. As I understand the effect you are trying to test, it is whether you see evidence of undue influence and then whether there is evidence that this is welfare reducing. You find some evidence that is consistent with undue influence--but is this really undue influence? I have a hard time thinking of \$30 to eat an insect as being of such a weight that one feels they cannot turn it down. It seems you are relying on the notion that if \$3 is absolutely unconvincing, and \$300,000 would be absolutely compelling, then any point between must represent a portion of this compelling effect. Is it not possible that there is a difference between what you are seeing and what would be observed with a truly compelling incentive?

3. Behavioral welfare is a useful tool and one that is well suited to this exercise. That said, it relies very heavily on specific assumptions about how choice relates to wellbeing. It seems like in a journal for general audiences you would need to spend significant ink explaining these underlying assumptions and how they could fail. These would be key in allowing general readership to understand and interpret that results with the same level of skepticism that an economist might.

Reviewer #2:

Remarks to the Author:

The author tests the existence of undue inducement -- that incentives may distort decision making (by clouding judgement) in a manner that reduces welfare. The idea of undue inducement is an argument against providing monetary incentives to providers in markets of items that are often regarded as particularly sensitive, or potentially catastrophic to providers, such as egg donations, organ donations, etc, while generating large benefits to receivers. The author tests the idea of undue inducement in laboratory experiments. The author also develops a theory to explore the relationship between incentives and information seeking, as well as distorted decision making.

The experiments are very cleverly, creatively and well designed, and the theory is informative. All in all, this is an excellent paper that provides novel insights generated by solid empirical analyses.

I have a few comments.

1. The author argues that the analysis of undue inducement is important due to the laws preventing monetary incentives in transactions of, say, egg donations and organs. However, the author entirely focuses the analysis on incentives given to the donor, and disregards the benefits to the receiver from the transaction. The benefits to receivers can arguably be very large. How might the benefits to receivers play a role in the existence of undue inducement? This seems worth mentioning in the paper.

2. Page 3, paragraph 1: the author mentions "questionable decision making" and that such is "widespread." It seems the author here hinges the argument on that some people make questionable decisions (given rational people cannot be harmed by incentives). Hence, what does "questionable" mean? People acting against their own self-interest, as defined by some meta preferences? People being boundedly rational? Or? Also, it seems meager to argue that such behavior is widespread, based on three references that seem to be quite arbitrary representatives of the "widespread" questionable

decision making.

3. In that same paragraph, the author mentions previous work on the existence of undue inducement, but never reports any of their results and whether the results in the current study aligns with, or contradicts, those reported in previous studies. It seems worth mentioning what other studies found and what methods were used.

4. Footnote 21: what tests were used? T-tests? Chi2?

5. Typos:

Page 9, paragraph 3: sentence reads weird in middle of paragraph.

Page 15, paragraph 2, sentence 2: delete "increases."

Page 25, last paragraph, first sentence: delete punctuation.

Reviewer #3:

Remarks to the Author:

My comments are attached.

Reviewer #4:

Remarks to the Author:

This paper reports on two experiments and some associated modeling aiming to test the hypothesis that offering individuals large incentives can result in "undue inducement." Undue inducement is understood here as the phenomenon whereby the individual agrees to do something which is on balance harmful to them because the inducement has distorted their decision-making. The author distinguishes between two claims which he believes constitute the "undue inducement hypothesis" (UIH):

UIH-positive. A behavioral claim: "incentives will cause participants to engage in biased information processing and motivated reasoning about the transaction." (p.1)

UIH-normative. A normative claim: "these changes [to reasoning] cause harm, and that incentives should therefore be capped." (p.1)

The experiments aim to test these claims. The results of the experiments, as well as the associated Bayesian model of reasoning support UIH-positive but not UIH-normative.

Overall, the experiments appear well-designed and the results are compelling. Some comments follow for the author's consideration.

Major comment

1. I think the paper would be improved by some more discussion of its target. Two points stand out with respect to the author's interpretation of "undue inducement."

The first concerns UIH-normative. The paper assumes that if undue inducement is problematic, it is in virtue of it causing people to make bad decisions (e.g., ones that lead to harm). This is one view of what undue inducement would be. Another is not about the outcome of the decisions, but the process. On this view, undue inducement would occur if an incentive undermined someone's decision-making capacity. The problem would then not be a matter of harm, but of consent. It would be helpful to hear what the relevance of the experimental results are to this alternative view of undue inducement: do they provide evidence that decision-making capacity is undermined or intact?

The second concerns UIH-positive. The interpretation of UIH-positive seems to concern the effects of incentives on information-seeking behavior: higher incentives will lead individuals to seek out information that favors the incentivized action. On this interpretation biased cognitive processing still involves the agent gaining more information that is relevant to a decision. But this is not, I would expect, what most people think it means for someone to engage in biased information processing and motivated reasoning. Most people would think that it means the person will act in some sense irrationally—for example, by putting greater weight on the evidence they already have that favors one choice over another, avoiding finding out negative information, or seeking confirmation of what one already believes. I think the author should explain why they chose the narrow interpretation of UIH-positive that they chose and whether the experimental results tell us anything about other senses of biased decision-making.

Minor comments

1. Note that as the author construes it UIH-normative actually comprises two claims:

The incentives cause harm.

The incentives should be scrapped (because they cause harm)

The paper addresses the first of these. This could be clarified in the text.

2. "Undue inducement concerns are also a frequent reason why research ethics boards intervene with designs of the social science studies they oversee, including those using payments of the order of magnitude I employ in my experiments." (p.2). Citation needed to back up this claim.

3. I'm not sure what "resonate" means on p.3. But the findings don't seem to imply that participants are discounting risks, since that would suggest that the participants have lowered their expectation of the costs associated with eating the insects.

4. "I then test UIH-normative by considering the family of welfare functions that vary the weight they place on the ten to twenty percent of subjects whose decision to participate was (ex-post) mistaken." (p.4). This was very hard to follow.

5. p.7. Subjects should be culturally situated, i.e., I take it they are from cultures where insects are not considered acceptable food items? Was this an exclusion criterion?

6. pp.8-9. It seems like a stretch to call this a "revealed" reservation price. Sounds more like they are asked to state their reservation price.

7. "Viewing a video may potentially alter reservation prices normatively invalid way." (p.9.) Missing "in a". Also, I didn't understand this claim.

8. p.12. What do you make of the high baseline preference for the encouraging video? Consider mentioning in the discussion.

9. Given the applied ethics audience that is likely to be most interested in these results and the fact that they are likely to raise some of the potential objections that the author considers but not in the same terms, I think it would be very valuable to have a plain English explanation of the results of the first experiment.

10. p.21 and elsewhere. As noted above, this is not the common language sense of "biased." More importantly, I think that the modeling here captures one way in which incentives may affect decision-making, which is that they make the individual more likely to gather information that favors or disfavors a specific choice. But it does not capture the notion that the incentive affects the decision-making without going via information acquisition, e.g., by causing immediate emotions to override rational decision-making.

11. p.38, line 7. Missing "be."

Author Rebuttal to Initial comments**Response to Reviewer 1:**

1. My first concern is whether the experiments are able to address the topic of interest. This is a substantive concern. The authors use a decision to consume insects as a stand in for decisions that would hold substantive risk of harm. My guess is that the subject pool is at least somewhat aware of efforts to incorporate insects in western diets and would not perceive this is holding some long term risk of harm. Without that, the risk they are mitigating is perhaps just a risk of disgust--perhaps a much smaller effect and one that may not be a good behavioral analog. In my read of the manuscript I felt like the authors glossed over this potential, which is honestly very key to whether the paper has merit. Is there some way to grapple with this?

I have added the following paragraph to the introduction to explain why the experiments are informative:

I investigate UIH in laboratory experiments. This is possible even with stakes on the order of dozens or hundreds of dollars for two reasons. First, as I will demonstrate, even these stakes cause the information-biasing UIH-positive predicts. The ethics literature views this as a key cause of UIH-normative. Second, as the ethics literature clarifies, an offer that is genuinely too good to refuse cannot constitute undue inducement. Undue inducement requires that a subject with undistorted judgment would refuse the offer, but that the incentive warps her judgment in a way that makes her accept it¹⁴. Therefore, undue inducement requires that an offer is high relative to the incentivized activity— but not so high that it is genuinely too good to refuse. I satisfy this requirement by choosing activities whose downsides are commensurate to the incentive amounts I employ.

I am referring to the following excerpt from reference [14]:

“It is important to appreciate that the distinction between an unproblematic (non-undue) inducement and an undue inducement is not a feature of the inducement itself. It is a function of the relation between the inducement and the subject’s response to it. As a general category, an inducement is undue only when it predictably triggers irrational decision-making given the agent’s own settled (and reasonable) values and aims. As the official IRB guidebook puts it, an offer is troublesome if it is so “attractive [that it] may blind prospective subjects to the risks or impair their ability to exercise proper judgment” about the risks of participation.¹³ Distortion of judgement is the key. An inducement is not morally problematic simply because it motivates the target to make a decision he or she would otherwise not make, if it is reasonable or in the recipient’s interest to accept the proposal. An offer is not problematic if it is genuinely too good to refuse. It is problematic if it seems to be too good to be refused and would be refused if the agent’s judgement were not blinded or clouded or impaired.”

Experiment 2 is designed such that I know exactly how high the incentive is relative to the risk (and thus what constitutes an offer that is genuinely too good to refuse). I find that UIH-normative is violated in this treatment, too.

Experiment 2 seeks to address your concern about long-term risks by introducing a treatment in which the potential downside from participation realizes only with a three-month delay. This treatment does not change any results. While a three-month delay is, of course, shorter than the time between, for instance, human egg donation and the potential onset of cancer, subjects in my experiment still discount the future payment heavily (they prefer 84 euros paid immediately over 100 euros paid with a three-month delay).

Generally, any laboratory behavioral experiment is a model of a real-world situation. It selects some elements to retain, and abstracts from others to hone in on a specific mechanism of interest.

The elements the current paper focuses on are, as stated in the introduction:

(i) subjects need to evaluate the non-monetary consequences of a transaction in terms of dollars and cents, ... (ii) transaction that is unfamiliar to subjects and allows for biased information acquisition.

In particular, point (ii) allows me to study the welfare effect of incentives through the information acquisition channel.

To better convey the boundary conditions of this paper, I have highlighted the consequentialist

approach in the introduction:

Interpreted consequentialistically⁵, two distinct claims make up the Undue Inducement Hypothesis

I have also added the following paragraph to Section 4, which highlights the mechanisms from which the current paper abstracts, but that could potentially play a role in real-world, larger-stakes transactions:

Future research should test the robustness of my findings in other contexts, with different subject pools, and with transactions currently subject to undue-inducement-based regulation, such as female egg donation or gestational surrogacy. Such cases might recruit psychological mechanisms not present in the current study, such as choking under pressure⁴⁴.

Importantly, stakes of huge magnitudes are not necessary to meaningfully study the welfare effects of incentives through informational channels.

[14] Alan Wertheimer and Franklin G Miller. Payment for research participation: a coercive offer? Journal of Medical Ethics, 34(5):389–392, 2008.

[44] Dan Ariely, Uri Gneezy, George Loewenstein, and Nina Mazar. Large stakes and big mistakes. The Review of Economic Studies, 76(2):451–469, 2009.

2. The subject pool appears to be university students are highly ranked US schools. This is a very common criticism of experimental economics, but perhaps uniquely important in this endeavor. The scale of rewards (up to \$30) are not likely to be so outside the everyday experience of this pool as to be compelling. As I understand the effect you are trying to test, it is whether you see evidence of undue influence and then whether there is evidence that this is welfare reducing. You find some evidence that is consistent with undue influence-- but is this really undue influence? I have a hard time thinking of \$30 to eat an insect as being of such a weight that one feels they cannot turn it down. It seems you are relying on the notion that if \$3 is absolutely unappealing, and \$300,000 would be absolutely compelling, then any point between must represent a portion of this compelling effect. Is it not possible that there is a difference between what you are seeing and what would be observed with a truly compelling incentive?

My response to these points refers to reference 14 cited in response to your previous question, according to which an offer that is genuinely too good to refuse cannot constitute undue inducement. The case of \$300,000 for eating insects appears unlikely to constitute undue inducement, simply because \$300,000 for eating two crickets produced for human consumption is genuinely too good to refuse (unless, for instance, one has a deadly allergy). The trick for designing an experiment on undue inducement is to find the incentive amount that may *appear* to be too good to refuse, but that is just *low enough* such that this appearance is mistaken. Experiment 2, therefore, explores a variety of incentive amounts. Gain $m = \$20, \$30, \$70, \80 for taking a gamble in which one can lose \$100 (but can always keep the incentive). In this case, it is obvious that using $m = 300,000$ would not make sense, because that offer would genuinely be too good to refuse. By the same argument, it would not make sense to offer $m = \$100$, which would simply transform the decision into a win 100 / win 0 gamble. If there were a reasonably broad range of incentive amounts for which undue inducement occurred, I would thus detect it in my experiment (in at least one of the multiple treatments I explore).

I agree that my experiment may not capture all relevant psychological mechanisms that may play a role in real-life undue inducement cases. I now highlight this point in Section 4, already cited above:

Future research should test the robustness of my findings in other contexts, with different subject pools, and with transactions currently subject to undue-inducement-based regulation, such as female egg donation or gestational surrogacy. Such cases might recruit psychological mechanisms not present in the current study, such as choking under pressure⁴⁴.

3. Behavioral welfare is a useful tool and one that is well suited to this exercise. That said, it relies very heavily on specific assumptions about how choice relates to wellbeing. It seems like in a journal for general audiences you would need to spend significant ink explaining these underlying assumptions and how they could fail. These would be key in allowing general readership to understand and interpret that results with the same level of skepticism that an economist might.

Next to the addition to Section 4 referenced in response to your previous question, I have added the following to the introduction

Interpreted consequentialistically⁵, two distinct claims make up the Undue Inducement Hypothesis (henceforth: UIH).

This paper empirically tests the cognitive underpinnings of the Undue Inducement Hypothesis.

Overall, this paper shows that incentives exert the behavioral effects the undue inducement literature hypothesizes. Yet, from a consequentialist point of view, these effects do not justify that literature's normative conclusions.

The middle sentence implies that the research does not address, for instance, affective underpinnings, if there are any. The paragraph of Section 4 cited in response to your previous question outlines what viewpoints the current paper abstracts from, including the affective mechanism of choking under pressure. It now also includes the following language:

To address non-consequentialist concepts such as autonomy and consent, future research should also extend the empirical study of undue inducement beyond the welfarist framework used here.

*[5] Walter Sinnott-Armstrong. Consequentialism. In Edward N. Zalta and Uri Nodelman, editors, *The Stanford Encyclopedia of Philosophy*. Metaphysics Research Lab, Stanford University, Winter 2022 edition, 2022.*

Response to Reviewer 2:

The experiments are very cleverly, creatively and well designed, and the theory is informative. All in all, this is an excellent paper that provides novel insights generated by solid empirical analyses.

Thank you, I much appreciate reading this!

I have a few comments.

1. The author argues that the analysis of undue inducement is important due to the laws preventing monetary incentives in transactions of, say, egg donations and organs. However, the author entirely focuses the analysis on incentives given to the donor, and disregards the benefits to the receiver from the transaction. The benefits to receivers can arguably be very large. How might the benefits to receivers play a role in the existence of undue inducement? This seems worth mentioning in the paper.

The second paragraph of the introduction now contains the following sentence:

Restrictions are well-advised if incentives really impede decision quality to an extent that outweighs the benefits of the increased payment (including, potentially, benefits to society and other parties).

I don't dwell on this point further, for two reasons. First, assume that the benefits of, say, organ donation to any recipient remain constant in the supply of organs. This might be an appropriate assumption if, for instance, organs are allocated in random priority order, as is currently the case in many countries. In this case, incentives do not interact with the benefits. Hence, accounting for the benefits presents a stronger argument in favor of permitting the transaction at any incentive, but it does not make UIH-normative more or less likely satisfied. (If the marginal benefits of provision to recipients are strongly decreasing in the supplied amount, then UIH-normative might more likely be satisfied.)

Second, unlike economists, the ethics literature is typically unwilling to trade off the utility of recipients with that of donors. Because this paper seeks to address ethicists, among others, I prefer not to base the analysis on assumptions they reject.

2. Page 3, paragraph 1: the author mentions "questionable decision making" and that such is "widespread." It seems the author here hinges the argument on that some people make questionable decisions (given rational people cannot be harmed by incentives). Hence, what does "questionable" mean? People acting against their own self-interest, as defined by some meta preferences? People being boundedly rational? Or? Also, it seems meager to argue that such behavior is widespread, based on three references that seem to be quite arbitrary representatives of the "widespread" questionable decision making.

You are raising deep questions that require a book-length treatment to be addressed properly. I have changed the reference to the Handbook of Behavioral Economics [7] which contains precise definitions of mistakes, and the philosophical foundations on which these definitions rely. The book comprehensively reviews the economics profession's collected knowledge about questionable decision-making with applications in all domains that economists typically study.

[7] Bernheim, B. Douglas, Stefano DellaVigna, David Laibson, (eds.) *Handbook of behavioral economics: Applications and foundations*. 2019.

To answer your specific question regarding the definition of questionable decision-making I adapt in the paper, I hope the following two paragraphs of the introduction address your concerns:

For a subject who accepted the transaction, the difference between the incentive she received and this reservation price reveals potential choice mistakes. If she committed to eating them for \$30 but finds, after inspecting them, that she would need at least \$45, has inflicted harm of $$(45 - 30) = 15 onto herself. Generally, I define ex-post welfare as the difference between the reservation price and the incentive if the subject accepted the transaction and zero otherwise.

I measure whether the decision was good given the information the subject had at the time, rather than in hindsight. Specifically, I calculate how confident a perfectly rational subject would have been that she will not lose the EUR100 after receiving the advisor's recommendation. Instead of asking subjects whether they wish to participate given the recommendation, I present subjects with a lottery in which the probability of not losing EUR100 equals the perfectly rational subject's confidence. I then measure what amount of money the real subject considers just as good as that lottery (her certainty equivalent). If a subject accepted the transaction in stage 1, but would rather lose a given sure amount of money than play the corresponding stage-2 lottery, her stage-1 decision was a mistake given the information she had at the time. The magnitude of her certainty equivalent measures the severity of the error.

Section 1 addresses concerns with the empirical implementation of the ex-post measure of welfare used in Experiment 1:

First, the incentive might directly affect ex-post reservation prices, for instance, through anchoring⁴². Second, the videos themselves may distort reservation prices, for example, because

they are tendentious. I simultaneously address these concerns by assuming that true reservation prices in each treatment follow the distribution revealed by subjects who could not watch a video and were given the \$3 incentive. I assign the subject with the highest reservation price in the (\$30, Video) treatment the highest reservation price observed in the (\$3, No Video) treatment. I perform a similar match for all other reservation price ranks. Based on these counterfactual reservation prices but using the subjects' actual participation decisions, I calculate counterfactual surplus. This measure incorporates the effect of incentives and information on participation decisions but excludes such effects for reservation prices.

3. In that same paragraph, the author mentions previous work on the existence of undue inducement, but never reports any of their results and whether the results in the current study aligns with, or contradicts, those reported in previous studies. It seems worth mentioning what other studies found and what methods were used.

I apologize for the omission. The introduction now contains the following language:

Direct empirical evidence on UIH is limited to a small number of case studies and unincentivized surveys concerning clinical trial participation⁸⁻¹¹. While these provide suggestive evidence that payments do not alter judgments about study risks, they neither study real choices nor perform formal welfare analysis.

4. Footnote 21: what tests were used? T-tests? Chi2?

I have added the following to Section 1:

Throughout, p-values reflect two-sided tests of hypotheses about coefficients in OLS regressions with standard errors clustered by subject, unless noted otherwise.

5. Typos:

Page 9, paragraph 3: sentence reads weird in middle of paragraph. Page 15, paragraph 2, sentence 2: delete "increases."

Page 25, last paragraph, first sentence: delete punctuation.

Thank you. Satisfying Nature Human Behavior's formatting restrictions required me to completely rewrite the entire paper.

Response to Reviewer #3:

The explanation of Experiment 2 could be clearer. To explain it, the paper states "After learning the incentive payment for the current round, the subject chooses one of two information structures, $I \in \{IG, IB\}$. Information structure IG is statewise biased towards G relative to B. The subject observes a stochastic signal from the chosen information

structure and then decides whether to participate in the gamble." Can this be written in plain English? I understand that the experiment aims to be close to the model developed in Section 3. But the reader should be able to understand the experiment without knowing the details of Section 3. When writing the summary, I tried to explain Experiment 2 in a couple of sentences and this was not straightforward. This is a problem. Readers should easily understand what is going on in Experiment 2.

Thank you for this comment. In rewriting the paper, I have placed great emphasis on making it as easy to read as possible. I have removed all technical language and unnecessary symbolism. Experiment 2 might have been hard to parse because of the references to the theoretical model. I have removed all such references. I have also moved an intuitive description of the model to Section 3, and the formal model itself into the SI.

In addition, I more clearly explain the contribution of Experiment 2 in the introduction:

Experiment 2 uses a broader range of incentive amounts, permits alternative welfare benchmarks, and explores moderators of the effect of incentives on decision quality.

The beginning of Section 2 explains further:

The previous analysis leaves open two questions. First, will the effect of participation incentives differ if we measure decision quality without the benefit of hindsight but based on the subject's

information at the time of decision? Second, are there cases in which UIH-normative is more likely satisfied, such as when incentives for a transaction are paid immediately, but potential downsides only realize with a delay?

Minor Comments

P8: It was unclear why some subjects watched the full video and other clips. This become clearer later in the paper. This could be make clearer on p8.

The third paragraph of Section 1 now explains:

To provide more continuous data about information preferences, subjects also select at least four of nine clips grouped in bins of three named “Reasons for eating insects”, “Reasons against eating insects”, and “Other information about eating insects.”

The reason why some subjects actually watch the clips rather than the video is simply to make the choice of clips potentially consequential. But because I am mostly interested in the effects of the videos (which provide less balanced information, and therefore more likely contribute to UIH-normative than a balanced selection of clips), I decided to assign only a very small fraction of subjects to watching the clips.

Figure 2 shows the results of experiment 1. The choice of the y-axis is a bit misleading. The left panel shows a proportion, but the y-axis is not from 0 to 1 but from 0.75 to 0.95. This makes the effect look bigger that it is. The same is true for Figure 5.

Given that the y-axis is clearly labeled in both these graphs, I would prefer not to deviate from the common practice of showing the part of the graph where things happen.

It would be good to discuss the effect size when discussing figure 2. Is a 7.5% difference a big difference?

The 7.5 %-points effect is big in the sense that it corresponds to 39.9% of the 18.8% of subjects who select the discouraging video when the incentive is low. The effect is small in the sense that it corresponds to 9.2% of the subjects who select the encouraging video when the incentive is low.

Another way to view the issue is through the lens of statistical power. If the switch to the encouraging video caused by the incentive were concentrated among a substantial fraction of individuals with high reservation prices (e.g. \$50) to accept the transaction, this would show up as a clear increase in harm in Figure 1.C.

Moreover, the effect on information choice in Experiment 2 is 16.1 %-points, and thus more than twice as large as in Experiment 1. UIH-positive does not make a claim about the magnitude of the effect one should expect. Therefore, and due to tightly limited space (5000 words), I have decided not to discuss these magnitudes aside from reporting them.

Figure 2 left panel shows a proportion (from 0 to 1), not a percentage (from 0 to 100) as the y-axis states.

Corrected.

One of the link to the video in Appendix E leads to a private video that cannot be watched.

Thank you for the comment. This was caused by a change in youtube policies. I have now included both videos in the replication materials on the Harvard dataverse such that they will continue to be available regardless of such policy changes.

Response to Reviewer #4:

1. I think the paper would be improved by some more discussion of its target. Two points stand out with respect to the author's interpretation of "undue inducement."

The first concerns UIH-normative. The paper assumes that if undue inducement is problematic, it is in virtue of it causing people to make bad decisions (e.g., ones that lead to harm). This is one view of what undue inducement would be. Another is not about the outcome of the decisions, but the process. On this view, undue inducement would occur if an incentive undermined someone's decision-making capacity. The problem would then not be a matter of harm, but of consent. It would be helpful to hear what the relevance of the experimental results are to this alternative view of undue inducement: do they provide evidence that decision-making capacity is undermined or intact?

I have changed the introduction to make clear that the paper relies on a consequentialist interpretation of undue inducement:

Interpreted consequentialistically⁵, two distinct claims make up the Undue Inducement Hypothesis (henceforth: UIH).

I re-emphasize this point in the discussion section:

To address non-consequentialist concepts such as autonomy and consent, future research should also extend the empirical study of undue inducement beyond the welfarist framework used here.

Based on the consequentialist interpretation I employ in this paper, any issues with undermining decision-making capacity (which undermine consent) are relevant only to the extent that they cause ex-ante or ex-post undesirable outcomes, according to the definitions of the outcome measures in the introduction. The sentence from the discussion section cited above highlights that non-consequentialist concerns with consent are not addressed by the current study.

I have included some of the evidence from Experiment 2 that speaks to the rationality of subjects through data other than the main measures of surplus, see Section 2:

As supporting evidence, each round of the first part also elicits subjective beliefs about whether the loss will materialize, after the subject has seen the advisor's recommendation. Rational behavior requires that mean beliefs do not depend on the incentive amount (law of iterated expectations).

If the incentive amount increases to EUR70 or more, mean subjective posteriors increase by 7.7 percentage points ($p < 0.05$), to 56.6% (exceeds 50% with $p < 0.01$). This effect is inconsistent with rational inference and suggests potentially harmful effects of higher incentives on welfare.

Please note, however, that this evidence does not imply the way in which subjects process information from a given source. The alternative interpretation is that different biases apply to different information sources. In this case, even if the processing of information from any given source is completely unaffected by incentives, the fact that the incentives induce a different selection of information sources can still explain the aggregate effect on Bayesian posteriors I document.

The second concerns UIH-positive. The interpretation of UIH-positive seems to concern the effects of incentives on information-seeking behavior: higher incentives will lead individuals to seek out information that favors the incentivized action. On this interpretation biased cognitive processing still involves the agent gaining more information that is relevant to a decision. But this is not, I would expect, what most people think it means for someone to engage in biased information processing and motivated reasoning.

Most people would think that it means the person will act in some sense irrationally—for example, by putting greater weight on the evidence they already have that favors one choice over another, avoiding finding out negative information, or seeking confirmation of what one already believes. I think the author should explain why they chose the narrow interpretation of UIH-positive that they chose and whether the experimental results tell us anything about other senses of biased decision-making.

With apologies, the data show and the theory predicts that higher incentives will cause agents to seek more information from sources expected to be encouraging while *also* seeking *less* information from sources expected to be discouraging. Specifically, in Experiment 1, subjects can only select either the encouraging video or the discouraging video. Therefore, the fact that subjects choose the encouraging video more often if the incentive is high mechanically implies that they choose the discouraging video less often (Figure 1.A, left panel). In addition, when I allow subjects to select as many clips as they want (but require that they select at least four), I find that subjects select significantly fewer con-clips if the incentive is high (Figure 1.A, right panel). Similarly, in Experiment 2, the Cautious and Bold advisor are exactly symmetrical. There is no information-theoretically valid interpretation according to which one of the two

advisors provides more information. In both these experiments, therefore, acquiring more information of one type goes along with acquiring less information of the other type, even when this is not enforced mechanically (in case of the clips). Enforcing the decrease of the other type of information makes it more likely that UIH-normative would be satisfied.

I do address the concern that higher incentives might cause subjects to acquire more of one type of information without acquiring less of the other type of information. Indeed this happens in Experiment 3, which, due to lack of space in the main article, I relegated to SI, section C. I briefly refer to this finding in the paragraph just before Section 3 of the main article:

A separate experiment (see SI) tests whether UIH-normative is more likely satisfied if subjects search through a large amount of information rather than observing a single recommendation from an advisor. There, too, choices satisfy UIH-positive but violate UIH-normative. In fact, higher incentives increase efforts to determine whether participation is the right choice.

I agree that depending on the literature a scholar usually moves in, my use of the term “bias” may seem completely natural or quite unusual. The reason is that different strands of literature use the term “bias” in different ways. This paper uses it in the way the literature on information acquisition and media bias does. See below for details. It is related to the use of this term in the statistics literature. My use of the term is also related to its use in parts of the cognitive psychology literature, such as the “positive testing bias”, which, in many instances, is simply a description of a behavioral tendency, rather than a deviation from rationality. I am aware, however, that my use of the term differs from the use in behavioral economics, where “bias” means “deviation from rationality.” As the model (now in section 3) shows, UIH-positive is a bias only in the sense of Gentzkow et al., but not necessarily in the sense of behavioral economics.

I attempted to address this issue in earlier versions of this paper by using the term “information skewing” rather than “information biasing,” but I have made the experience that this confuses readers, especially those familiar with the literature on information acquisition and media bias.

Regarding the details of the definition of the term that I employ, here is the formal definition (from Section 2 of Gentzkow et al., 2015):

Suppose that there is an unobserved state of the world $\theta \in \{L, R\}$, whose values we associate with outcomes favorable to the left and right sides of a one-dimensional political spectrum, respectively. Define the raw facts gathered by a news outlet to be a (possibly high-dimensional) random variable $s \in S$ whose distribution depends on θ , and define a news report by $n \in N$. A reporting strategy σ is a possibly stochastic mapping from S to N .

We define bias as a partial order on reporting strategies. We say a strategy σ is *biased to the right (left) of σ'* if, loosely speaking, a consumer who believed a firm’s strategy was σ' would tend to shift her beliefs to the right (left) if the firm deviated to σ .

More precisely, let $\mu(n|\sigma)$ be a Bayesian consumer’s posterior probability that the state is R when she observes n from a firm believed to be playing strategy σ . Let $\lambda(\tilde{\sigma}|\sigma)$ be the distribution of μ when a consumer believes a firm is playing σ and it actually plays $\tilde{\sigma}$. Say two reporting strategies σ and σ' are *consistent* if for each of them n has the same support (i.e., the set of n that can be reported is the same), and they preserve the relative meaning

In a previous version of the paper, I explained formally how this definition specializes to the one I use in my setting. I copy that explanation here:

³⁶The following argument shows the equivalence to the definition stated in Gentzkow et al. (2015) (the definition of statewise bias does not appear in Gentzkow et al. (2015)). Consider two information structures I and I' . Let $\lambda(I|I')$ be the distribution of an observers' posterior probability that $s = G$ if he believes that the report is generated by information structure I even though the report is actually generated by information structure I' . Gentzkow et al. (2015) define I' as *biased towards G* relative to I if $\lambda(I|I')$ first-order stochastically dominates $\lambda(I|I)$. An agent who believes the signal is generated by information structure I will have a posterior belief that $s = G$ of $\gamma_{\mathcal{G},I} = \frac{\mu p_G}{\mu p_G + (1-\mu)p_B}$ if he observes \mathcal{G} and $\gamma_{\mathcal{B},I} = \frac{\mu(1-p_G)}{\mu(1-p_G) + (1-\mu)(1-p_B)}$ if he observes \mathcal{B} . If information structure I actually generates these signals, then the agent's posterior will equal $\gamma_{\mathcal{G}}$ with probability $p = \mu p_G + (1-\mu)p_B$ and $\gamma_{\mathcal{B}}$ otherwise. If the signal is actually generated by information structure I' , then the agent's posterior will equal $\gamma_{\mathcal{G}}$ with probability $p' = \mu p'_G + (1-\mu)p'_B$ and $\gamma_{\mathcal{B}}$ otherwise. Because the support of the distribution of the agent's posterior probability is the same across these two cases, the required first order stochastic dominance relationship obtains if and only if p' exceeds p . Accordingly, I' is biased towards \mathcal{G} relative to I if and only if $p' \geq p$.

Gentzkow, Matthew, Jesse M Shapiro, and Daniel F Stone, "Media bias in the marketplace: Theory," in "Handbook of media economics," Vol. 1, Elsevier, 2015, pp. 623–645.

Minor comments

1. Note that as the author construes it UIH-normative actually comprises two claims: The incentives cause harm.

The incentives should be scrapped (because they cause harm)

The paper addresses the first of these. This could be clarified in the text.

The introduction of the paper defines UIH-normative as follows:

The normative part (henceforth: UIH-normative) is a welfare judgment. It claims that these changes cause harm and that some transactions are therefore acceptable only at low but not at high incentives.

Here, "these changes" refer to UIH-positive.

I define UIH-normative formally in Section 1:

Therefore, the higher incentive is no longer unambiguously preferable. Yet, the increase in losses is not per se sufficient for UIH-normative. The reason is that UIH-normative requires two conditions to hold simultaneously. First, the weight on losses must be sufficiently high to render preventing the transaction at the high incentive optimal (total welfare at the high incentive is lower than that from preventing the transaction). Second, the weight on losses must be sufficiently low to make participating at the low incentive acceptable (total welfare at the low incentive exceeds that from preventing the transaction).

A common interpretation of utilitarianism is that one ought to do the things that provide the highest societal utility (see, e.g. Section 6 in Sinnott-Armstrong, 2022)---though it is not the only interpretation. Because this definition entails that “total welfare is lower/higher than that from preventing the transaction,” this common interpretation of utilitarianism implies the claim that “if and only if total welfare at the low incentive exceeds that from preventing the transaction and total welfare at the high incentive falls short of that from preventing the transaction, the transaction should be allowed only at the low but not high incentive. Therefore, I see your point that “The paper addresses the first of these” when viewed according to some specific interpretations of utilitarianism. Nonetheless, I would prefer to stick with the interpretation common in economics, according to which harm implies that the harm-causing activity should be stopped.

Sinnott-Armstrong, Walter, "Consequentialism", *The Stanford Encyclopedia of Philosophy* (Winter 2022 Edition), Edward N. Zalta & Uri Nodelman (eds.), <https://plato.stanford.edu/archives/win2022/entries/consequentialism/>.

2. “Undue inducement concerns are also a frequent reason why research ethics boards intervene with designs of the social science studies they oversee, including those using payments of the order of magnitude I employ in my experiments.” (p.2). Citation needed to back up this claim.

That statement was based on my own experience and based on conversations with other

experimental economists. Unfortunately, I do not have written documentation. I have therefore dropped this claim. In its place, I have added an explanation of why it is meaningfully possible to study undue inducement concerns using the stakes I employ in my experiments:

I investigate UIH in laboratory experiments. This is possible even with stakes on the order of dozens or hundreds of dollars for two reasons. First, as I will demonstrate, even these stakes cause the information-biasing UIH-positive predicts. The ethics literature views this as a key cause of UIH-normative. Second, as the ethics literature clarifies, an offer that is genuinely too good to refuse cannot constitute undue inducement. Undue inducement requires that a subject with undistorted judgment would refuse the offer, but that the incentive warps her judgment in a way that makes her accept it¹⁴. Therefore, undue inducement requires that an offer is high relative to the incentivized activity— but not so high that it is genuinely too good to refuse. I satisfy this requirement by choosing activities whose downsides are commensurate to the incentive amounts I employ.

3. I'm not sure what "resonate" means on p.3. But the findings don't seem to imply that participants are discounting risks, since that would suggest that the participants have lowered their expectation of the costs associated with eating the insects.

I have replaced the offending language as follows:

These findings dovetail with the idea that “payments lead donors to discount risks”³.

There are two possible interpretations of the term “discount risks” (on which the source document by the American Society for Reproductive Medicine provides no guidance). The first, which I implicitly adopt in this paper, is that the subject would discount risks in the sense of being less willing to learn about them. The second, which you might be referring to, is that the subject is aware of a risk, but begins weighing that risk differently. If it is the case that incentives cause changes in the expectations of the costs associated with eating the insects, then the main analysis of Experiment 1 is invalid because the ex-post reservation prices would not measure “true” preferences but preferences that reflect invalidly altered expectations. Section 1 addresses this concern as follows:

There are two potential concerns about this analysis. First, the incentive might directly affect ex-post reservation prices, for instance, through anchoring⁴². Second, the videos themselves may distort reservation prices, for example, because they are tendentious. I simultaneously address these concerns by assuming that true reservation prices in each treatment follow the distribution revealed by subjects who could not watch a video and were given the \$3 incentive. I assign the subject with the highest reservation price in the (\$30, Video) treatment the highest reservation price observed in the (\$3, No Video) treatment. I perform a similar match for all other reservation price ranks. Based on these counterfactual reservation prices but using the subjects' actual participation decisions, I calculate counterfactual surplus. This measure incorporates the effect of incentives and information on participation decisions but excludes such effects for reservation prices.

4. “I then test UIH-normative by considering the family of welfare functions that vary the weight they place on the ten to twenty percent of subjects whose decision to participate was (ex-post) mistaken.” (p.4). This was very hard to follow.

I have replaced the offending sentence as follows:

Formal tests of UIH-normative calculate welfare by aggregating surplus across subjects. I consider social planners that vary in the weight they place on losses incurred from participation. Even though ten to twenty percent of subjects are harmed by their decision to participate, I find that a planner would never want to allow the transaction at the low incentive but prevent it at the high incentive, for any weight on losses—contrary to UIH-normative. (A planner who weighs losses heavily prevents the transaction altogether.)

Section 1 explains in more detail:

UIH-normative requires two conditions to hold simultaneously. First, the weight on losses must be sufficiently high to render preventing the transaction at the high incentive optimal (total welfare at the high incentive is lower than that from preventing the transaction). Second, the weight on losses must be sufficiently low to make participating at the low incentive acceptable (total welfare at the low incentive exceeds that from preventing the transaction). Amongst the class of welfare functions that place weight α on losses and weight $(1 - \alpha)$ on gains, ...

5. p.7. Subjects should be culturally situated, i.e., I take it they are from cultures where insects are not considered acceptable food items? Was this an exclusion criterion?

No, ethnicity was not an exclusion criterion. Invitation emails mention that the experiment will involve the consumption of food items on the spot, and ask recipients not to participate if they have food allergies, are vegetarian or vegan, or eat kosher or halal.

Importantly, subjects are randomly assigned to treatments. If a substantial fraction of subjects were happy eating insects at any price, this would therefore not affect any treatment effects that arise from varying the incentive amount.

I have data on ethnicity, and I have checked whether ethnicity predicts reservation prices in my study. I have included the corresponding analysis in SI A.2, copied here for your convenience:

Table A.3: Effect of subjects' ethnicity on participation and reservation prices

VARIABLES	(1)	(2)	(3)
	Offer accepted	Reservation price	
		before handout	after handout
Ethnicity			
Black	0.035 (0.067)	2.098 (4.633)	1.957 (4.982)
Asian	-0.061 (0.039)	5.538* (2.955)	3.247 (3.189)
Hispanic	0.083 (0.070)	-2.945 (4.898)	-4.300 (4.993)
Indian	-0.123 (0.081)	15.375** (6.529)	17.552** (7.062)
Other	0.005 (0.074)	1.059 (5.170)	2.190 (5.988)
High incentive	0.283*** (0.032)		
Constant	0.386*** (0.028)	24.865*** (1.557)	28.988*** (1.738)
Observations	6,552	3,276	3,276
Subjects	671	671	671

Notes: Column 1 estimated by OLS. Columns 2 and 3 estimated with interval regression. White is the omitted category. Standard errors in parentheses, clustered by subject.

I only observe that Indians (4% of the sample) have significantly and substantially elevated reservation prices relative to the omitted category of white Americans (56% of the sample). There are no significantly lower reservation prices for Asians (22% of the sample) and Hispanics (6% of the sample). If anything they are slightly higher for Asians. One possible reason is that within broad cultural categories such as Asians or Hispanics, entomophagy is often concentrated amongst relatively small subpopulations (my Chinese PhD students inform me, for instance, that entomophagy is only practiced in a few regions in the south of China, and Mexicans make up only a relatively small part of Hispanics). A second possible reason is that I use five species of insects. Of the cultures that eat insects, many only eat one or two but still consider the others disgusting. For instance, while chapulines (grasshoppers) are typical in some Mexican cuisines

(mainly Oaxaca), Mexican cuisine appears not to use silkworm pupae or mole crickets both of which look and smell very different from chapulines.

Appendix Table A.1 shows the ethnic makeup of the subject population in each treatment, along with other demographics.

Table A.1: Summary statistics and randomization check.

Treatment condition					
Incentive	\$30	\$3	\$30	\$3	
Video	Yes	Yes	No	No	
Variable	Mean				p -value ^a
Male	0.55	0.53	0.54	0.54	1.00
Age	21.43	22.01	21.37	21.30	0.34
Ethnicity					
African-American	0.05	0.06	0.07	0.07	0.70
Caucasian	0.57	0.51	0.59	0.56	0.28
East Asian	0.19	0.26	0.19	0.23	0.22
Hispanic	0.07	0.08	0.04	0.04	0.99
Indian	0.03	0.04	0.04	0.07	0.56
Other	0.08	0.05	0.07	0.04	0.70
Monthly spending in USD	251.72	301.40	289.07	288.42	0.44
Year of study ^b	3.50	3.60	3.61	3.47	0.32
Graduate student	0.13	0.15	0.13	0.05	0.07
Field of study					
Arts and humanities	0.16	0.09	0.13	0.11	0.04
Business or economics	0.27	0.36	0.34	0.43	0.09
Engineering	0.20	0.16	0.11	0.12	0.49
Science	0.21	0.23	0.27	0.23	0.47
Social science (excluding business and economics)	0.17	0.17	0.15	0.11	0.59
Political orientation ^c	0.50	0.32	0.27	0.09	0.08
Raven's score ^d	14.77	14.76	14.69	14.68	1.00
CRT score ^e	3.76	3.80	3.50	3.22	0.08
Experience with insects as food (1 = Yes, 0 = No)					
Has intentionally eaten insects before	0.19	0.22	0.19	0.20	0.71
Grown up in culture that practices entomophagy	0.15	0.14	0.13	0.15	0.92
Grown up eating mostly western foods	0.81	0.73	0.82	0.78	0.06
Had a pet that feeds on store-bought insects	0.25	0.25	0.21	0.26	0.68
Knew that this study concerns insect eating	0.20	0.30	0.26	0.29	0.23

^a*p*-value of the test of joint significance of a regression of the indicated variable on treatment dummies.

^bYear of study only includes undergraduate students.

^cPolitical orientation is measured on a scale of -2 (conservative) to 2 (liberal).

^dRaven's score is measured on a scale of 0 to 24.

^eCRT score indicates the number of correct answers (out of 6) on the Toplak et al.⁴⁵ test.

6. pp.8-9. It seems like a stretch to call this a "revealed" reservation price. Sounds more like they are asked to state their reservation price.

In economics, any decision that is carried out with a positive probability is said to reveal preferences. Any statement that will certainly not have a consumption consequence is called a stated preference. The multiple price lists I use to elicit reservation prices have a positive probability of determining the subjects' consumption (this method is one of the most standard revealed preference methods used in experimental economics, see, e.g. the highly cited paper by Andersen et al., 2006). Hence, from the point of view of experimental economics, it would be incorrect to say that subjects "are asked to state their reservation price."

Andersen, S., Harrison, G. W., Lau, M. I., & Rutström, E. E. (2006). Elicitation using multiple price list formats. *Experimental Economics*, 9, 383-405.

7. "Viewing a video may potentially alter reservation prices normatively invalid way." (p.9.) Missing "in a". Also, I didn't understand this claim.

I have replaced the offending sentence with the following:

the videos themselves may distort reservation prices, for example, because they are tendentious

8. p.12. What do you make of the high baseline preference for the encouraging video? Consider mentioning in the discussion.

My interpretation is that many subjects already know why they might not want to eat insects. Importantly, in Experiment 2, the baseline preference for the Bold Advisor (which corresponds to the encouraging video in Experiment 1) is only around 1/3. The fact that I find similar results across the two experiments suggests that the high baseline preference for the encouraging video in Experiment 1 does not have a material effect on the findings. Given the tight word limit, I have chosen not to include this argument in the discussion section. But I am happy to change this on request.

9. Given the applied ethics audience that is likely to be most interested in these results and the fact that they are likely to raise some of the potential objections that the author considers but not in the same terms, I think it would be very valuable to have a plain English

explanation of the results of the first experiment.

I have rewritten the entire paper to make it much more accessible to audiences outside of economics. The introduction now summarizes the main conclusion of Experiment 1 as follows:

Even though ten to twenty percent of subjects are harmed by their decision to participate, I find that a planner would never want to allow the transaction at the low incentive but prevent it at the high incentive, for any weight on losses—contrary to UIH-normative.

10. p.21 and elsewhere. As noted above, this is not the common language sense of “biased.” More importantly, I think that the modeling here captures one way in which incentives may affect decision-making, which is that they make the individual more likely to gather information that favors or disfavors a specific choice. But it does not capture the notion

that the incentive affects the decision-making without going via information acquisition, e.g., by causing immediate emotions to override rational decision-making.

Please see my answer to your earlier point above. In my experience, the way in which scholars of different disciplines understand the term bias is closely linked to the way in which their field defines it, which varies across fields.

Decision Letter, first revision:

Our ref: NATHUMBEHAV-23010047A

8th September 2023

Dear Dr. Ambuehl,

Thank you for your patience as we’ve prepared the guidelines for final submission of your Nature Human Behaviour manuscript, "Can Incentives Cause Harm? Tests of Undue Inducement" (NATHUMBEHAV-23010047A). Please carefully follow the step-by-step instructions provided in the attached file, and add a response in each row of the table to indicate the changes that you have made. Please also address the additional marked-up edits we have proposed within the reporting summary. Ensuring that each point is addressed will help to ensure that your revised manuscript can be swiftly

handed over to our production team.

We would hope to receive your revised paper, with all of the requested files and forms within two-three weeks. Please get in contact with us if you anticipate delays.

Nature Human Behaviour offers a Transparent Peer Review option for new original research manuscripts submitted after December 1st, 2019. As part of this initiative, we encourage our authors to support increased transparency into the peer review process by agreeing to have the reviewer comments, author rebuttal letters, and editorial decision letters published as a Supplementary item. When you submit your final files please clearly state in your cover letter whether or not you would like to participate in this initiative. Please note that failure to state your preference will result in delays in accepting your manuscript for publication.

In recognition of the time and expertise our reviewers provide to Nature Human Behaviour's editorial process, we would like to formally acknowledge their contribution to the external peer review of your manuscript entitled "Can Incentives Cause Harm? Tests of Undue Inducement". For those reviewers who give their assent, we will be publishing their names alongside the published article.

Cover suggestions

We welcome submissions of artwork for consideration for our cover. For more information, please see our https://www.nature.com/documents/Nature_covers_author_guide.pdf target="new"> guide for cover artwork.

ORCID

Non-corresponding authors do not have to link their ORCIDs but are encouraged to do so. Please note that it will not be possible to add/modify ORCIDs at proof. Thus, please let your co-authors know that if they wish to have their ORCID added to the paper they must follow the procedure described in the following link prior to acceptance:

Nature Human Behaviour has now transitioned to a unified Rights Collection system which will allow our Author Services team to quickly and easily collect the rights and permissions required to publish your work. Approximately 10 days after your paper is formally accepted, you will receive an email in providing you with a link to complete the grant of rights. If your paper is eligible for Open Access, our Author Services team will also be in touch regarding any additional information that may be required to arrange payment for your article.

Please note that *Nature Human Behaviour* is a Transformative Journal (TJ). Authors may publish their research with us through the traditional subscription access route or make their paper immediately open access through payment of an article-processing charge (APC). Authors will not be required to make a final decision about access to their article until it has been accepted. Find out more about Transformative Journals

[REDACTED]

Best regards,
Alex McKay
Editorial Assistant
Nature Human Behaviour

On behalf of

Jamie

Dr Jamie Horder
Senior Editor
Nature Human Behaviour

Reviewer #1:

Remarks to the Author:

I thank the authors for their thorough rewrite of much of the paper, and their work to address my prior concerns. I do have one remaining concern which I consider to be both addressable but important. The authors find a welfare effect of UIH, but that it is of a size that should not concern a policymaker using your chosen welfare measure. I continue to have a concern that the size of the welfare effect may be due to the relatively mild activity induced in both studies and the relatively mundane incentive scales. The authors address this in the conclusion by suggesting the possibility of additional biases in other contexts and the need to explore those contexts. I would be much more comfortable if the discussion framed the results more narrowly recognizing the limits of external validity in the welfare argument. The welfare effect could be much larger depending on how large the UIH-positive effect is. I worry the tone is a little too confident in the current results and may be taken out of context by some readers.

Reviewer #2:

Remarks to the Author:

I believe the author has addressed the comments to a satisfactory degree, and commend the author on an interesting and important paper.

Reviewer #3:

None

Reviewer #4:

Remarks to the Author:

Thanks to the author for the thorough responses. For the most part these are satisfactory and the paper is much improved. I have a couple of further comments in response to the author's rebuttal and subsequent changes to the paper. These all relate to my original comments as Reviewer #4.

1. Regarding the conception of UIH-normative. The author prefers to interpret "undue inducement" in terms of welfare, not in terms of the undermining of decision-making capacity. This is fine, as far as it goes. I would recommend that the addition of "Interpreted consequentialistically," be deleted. This does not add to understanding. More important that it is made clear in the discussion that the experiments do not tell us anything about the impact of inducements on decision-making capacity.

2. On the use of the term "bias." The author points out that analytically, if someone chooses an encouraging video then they do not choose a discouraging video. This misses the point of the original comment. The point is that either way the person gets more actual information that is relevant to their choice. But, again, the standard understanding of "biased" information processing and motivated reasoning is that it is *not rational*. The experiment does not have a way for the participants to be less rational because it always gives them more genuine information that is in fact relevant to their decision. There is a challenge in interdisciplinary work when the same term is used in different ways. In my view, any time there is a non-technical use for the term, it is very important to flag clearly (1) that the term is being used in a technical way that differs from the non-technical meaning, and (2) what the implications are with respect to the non-technical use of the term. If this is not made very clear then there is a danger that the results of the paper will be picked up by others in a way that is misleading.

3. In the previous comments, I noted that UIH-normative comprises two claims, but that the paper only addresses the first. The author has edited the introduction but not resolved the issue. The introduction states:

"The normative part (henceforth: UIH-normative) is a welfare judgment. It claims that these changes cause harm and that some transactions are therefore acceptable only at low but not at high incentives."

As defined here, UIH-normative still comprises two claims. Only the first is a "welfare judgment." This first claim is what the paper addresses. The second claim – that some transactions are unacceptable because harmful – is not addressed by the paper and does not need to be addressed by the paper. I think the paper should clarify this point: note that UIH-normative comprises two claims and that the paper is only addressing the first.

4. On culturally situating subjects. Ethnicity is not the point. The experiment aims to test the effects of incentives on an aversive experience. Someone who regarded eating insects as normal would not find it aversive. The author appears to have collected data on experience with insects as food. An explanation should be added for why this was not an exclusion criterion.

Author Rebuttal, first revision:

Response to R1

"The welfare effect could be much larger depending on how large the UIH-positive effect is. I worry the tone is a little too confident in the current results and may be taken out of context by some readers."

I altered the previous sentence "*Given the potentially high costs of preventing voluntary transactions, the rules and guidelines restricting incentives due to undue inducement concerns should thus be re-examined.*" It now reads:

Given the potentially high costs of preventing voluntary transactions, experiments paralleling these reported here should be conducted in the field. Unless their results differ drastically from the current ones, the rules and guidelines restricting incentives due to undue inducement concerns should be reconsidered.

I do not make the connection between the magnitudes of UIH-positive and UIH-normative that R1 suggests because the two are logically independent, as the paper demonstrates. While it is true that UIH-positive could arise due to both rational or irrational behavior, there is simply no data (neither in my paper nor elsewhere) that suggests that such irrationality would be larger or smaller in field settings than in my current experiments.

Response to R4

I would recommend that the addition of “Interpreted consequentialistically,” be deleted.

I have deleted "*interpreted consequentialistically*."

"More important that it is made clear in the discussion that the experiments do not tell us anything about the impact of inducements on decision-making capacity."

The experiments tell us precisely that the impact on decision-making *quality*, when evaluated in terms of money-metric welfare, is at most as large as the increase in money-metric welfare due to the additional money payment, for the average subject. Formally, this is easy to see considering the case of $\alpha = \frac{1}{2}$ within Tables 1 and 2. $\alpha = \frac{1}{2}$ corresponds to utilitarian welfare, so it weights the gains and losses equally. If, for the average participant, the impact on decision-making quality evaluated in terms of money-metric welfare, were larger than the increase in money-metric welfare due to the additional money payment, then UIH-normative would be satisfied for $\alpha = \frac{1}{2}$. Tables 1 and 2 show that UIH-normative is consistently violated $\alpha = \frac{1}{2}$.

My previous response concerns decision-making *quality*. R4 uses the word decision-making *capacity* that he or she connects to the idea of consent. Consent is a non-consequentialist consideration and, hence, is outside of the domain of this paper. The paper repeatedly highlights that it exclusively focuses on consequentialist interpretations, for instance in the introduction and discussion sections:

Yet, from a consequentialist point of view, these effects do not justify that literature's normative conclusions.

To address non-consequentialist concepts such as autonomy and consent, future research should also extend the empirical study of undue inducement beyond the welfarist framework used here.

The standard understanding of “biased” information processing and motivated reasoning is that it is *not rational*. The experiment does not have a way for the participants to be less rational because it always gives them more genuine information that is in fact relevant to their decision

The assertion that the “experiment does not have a way for the participants to be less rational” is false. Briefly, a sufficiently risk-averse but rational subject will always seek the source of information that minimizes false positives. Neither the encouraging video in Experiment 1 nor the Bold Advisor in Experiment 2 minimize false positives, but the respective other information source does. Hence, if raising the incentive induces a sufficiently risk-averse subject to switch from the discouraging to the encouraging video, or from the Cautious to the Bold advisor, the incentive has induced the subject to choose a source that provides them with *less* information relevant to their decision. (Formally, this argument is easy to see in the context of Experiment 2 for the case of piecewise linear prospect theory preferences. I will provide the formal derivations on request.)

Two points that may further aid the interpretation of the term “biased”:

- There are two ways to use the words “biased” and “information” in combination. The first is to say that a given piece of information is biased. No notion of rationality is applicable here because no decision-maker is involved, regardless of whether one considers the technical or some vernacular use of the term. The information sources I use in the experiments are biased in this way.
- The second is to say that a decision-maker processes a given piece of information in a biased way. In this case, a notion of rationality is applicable, most commonly that of Bayesian updating. SI figure B.1, panels C and D, contain evidence for irrational information processing in Experiment 2. Bayes consistency (a.k.a. the law of iterated expectations) requires that the blue and red curves in each of these panels coincide. The fact that they do not shows that the expected posterior does not equal the prior, which is inconsistent with Bayesian rationality.

As defined here, UIH-normative still comprises two claims. Only the first is a “welfare judgment.” This first claim is what the paper addresses. The second claim – that some transactions are unacceptable because harmful – is not addressed by the paper and does not need to be addressed by the paper. I think the paper should clarify this point: note that UIH-normative comprises two claims and that the paper is only addressing the first.

I have clarified the two-part nature of the claim as follows:

“The normative part (henceforth: UIH-normative) is the composite claim that these changes cause harm and that some transactions are therefore acceptable only at low but not at high incentives.”

In light of the subsequent sentence “*This paper empirically tests...*” I think it is unlikely that readers will expect an investigation of the question of whether transactions that cause harm should be prevented.

4. "On culturally situating subjects. Ethnicity is not the point. The experiment aims to test the effects of incentives on an aversive experience. Someone who regarded eating insects as normal would not find it aversive. The author appears to have collected data on experience with insects as food. An explanation should be added for why this was not an exclusion criterion."

It is true that a key assumption on which the interpretation of my results relies is that subjects find eating insects aversive. The reservation prices I elicit measure whether they do. Any subject who does not will reveal reservation price 0. Previous experience with eating insects is merely a noisy predictor of that reservation price, which is why I did not use it as an exclusion criterion (some people may have experience because they positively like eating insects whereas others may have tried some species once and never again).

While I needed to cut the table showing the fraction of subjects with reservation price 0, they can be eyeballed off of Figure 2.C, as it is simply the fraction of subjects for whom welfare in the \$3 treatment is \$3 and for whom welfare in the \$30 treatment is \$30. In fact, there is no species for which more than 5% of subjects have a zero reservation price.

	(1)	(2)	(3)
	Reservation price		
	Fraction \leq \$0	Median	Fraction \geq \$60
2 house crickets	0.96	9.00	0.18
5 large mealworms	0.96	18.75	0.30
3 silkworm pupae	0.95	13.75	0.23
2 mole crickets	0.96	13.75	0.24
2 field crickets	0.95	13.75	0.22

Zero reservation prices are a concern to the extent that subjects could not express negative reservation prices (which correspond to a preference *for* eating insects). In order to overturn the results concerning UIH-normative, one needs to assume that subjects with that preference value eating insects implausibly much (by an order of magnitude more than the price at which the insects can be purchased on the internet). The final paragraph of SI A.3 derives the required magnitude.

Final Decision Letter:

Dear Professor Ambuehl,

We are pleased to inform you that your Article "An experimental test of whether financial incentives constitute undue inducement in decision-making", has now been accepted for publication in *Nature Human Behaviour*.

Please note that *Nature Human Behaviour* is a Transformative Journal (TJ). Authors may publish their research with us through the traditional subscription access route or make their paper immediately open access through payment of an article-processing charge (APC). Authors will not be required to make a final decision about access to their article until it has been accepted. Find out more about Transformative Journals

Acceptance of your manuscript is conditional on all authors' agreement with our publication policies (see <http://www.nature.com/nathumbehav/info/gta>). In particular your manuscript must not be published

elsewhere and there must be no announcement of the work to any media outlet until the publication date (the day on which it is uploaded onto our web site).

With best regards,
Jamie

Dr Jamie Horder
Senior Editor
Nature Human Behaviour